# Keeping an Eye on LLM Unlearning: The Hidden Risk and Remedy

**Jie Ren[1], Zhenwei Dai[2], Xianfeng Tang[2], Yue Xing[1], Shenglai Zeng[1], Hui Liu[2],**
**Jingying Zeng[2], Qiankun Peng[2], Samarth Varshney[2], Suhang Wang[3], Qi He[2],**
**Charu C. Aggarwal[4], Hui Liu[1]**
[1]Michigan State University, [2]Amazon, [3]The Pennsylvania State University,
[4]IBM T. J. Watson Research Center
{renjie3,xingyue1,zengshe1,liuhui7}@msu.edu
{zwdai, xianft, liunhu, zejingyi, qiankunp, varshsam, qih}@amazon.com
szw494@psu.edu, charu@us.ibm.com

## Abstract

Although Large Language Models (LLMs) have demonstrated impressive capabilities across a wide range of tasks, growing concerns have emerged over the misuse of sensitive, copyrighted, or harmful data during training. To address these concerns, unlearning techniques have been developed to remove the influence of specific data without retraining from scratch. However, this paper reveals a critical vulnerability in fine-tuning-based unlearning: a malicious user can craft a manipulated forgetting request that stealthily degrades the model's utility for benign users. We demonstrate this risk through a red-teaming *Stealthy Attack (SA)*, which is inspired by two key limitations of existing unlearning—the inability to constrain the scope of unlearning effect and the failure to distinguish benign tokens from unlearning signals. Prior work has shown that unlearned models tend to memorize forgetting data as unlearning signals, and respond with hallucinations or feigned ignorance when unlearning signals appear in the input. By subtly increasing the presence of common benign tokens in the forgetting data, SA enhances the connection between benign tokens and unlearning signals. As a result, when normal users include such tokens in their prompts, the model exhibits unlearning behaviors, leading to unintended utility degradation. To address this vulnerability, we propose *Scope-aware Unlearning (SU)*, a lightweight enhancement that introduces a scope term into the unlearning objective, encouraging the model to localize the forgetting effect. Our method requires no additional data processing, integrates seamlessly with existing fine-tuning frameworks, and significantly improves robustness against SA. Extensive experiments validate the effectiveness of both SA and SU. Our code is available at github.com/renjie3/sa_su.

## 1 Introduction

Large Language Models (LLMs) have demonstrated remarkable capabilities across a wide range of tasks, such as question answering [1], machine translation [2], text summarization [3], and dialogue generation [4]. This rapid advancement has been largely driven by training on massive datasets. However, growing concerns have been raised about the potential misuse of undesirable data in training, such as copyrighted materials, privacy-sensitive information, or harmful content [5, 6]. For instance, New York Times sued OpenAI and Microsoft for using its articles for training[1]. Under the

---

[1]https://www.nytimes.com/2023/12/27/business/media/new-york-times-open-ai-microsoft-lawsuit.html

39th Conference on Neural Information Processing Systems (NeurIPS 2025).

General Data Protection Regulation (GDPR) [7], the data is protected, and data owners—like The New York Times—retain the "right to be forgotten"[8].

To address these concerns, unlearning techniques have been proposed to remove the influence of specific data from large language models (LLMs) without retraining them from scratch [9, 10, 11, 12, 13, 14]. Specifically, model providers who develop and deploy LLM can offer unlearning services that accept user-submitted data removal requests. Upon receiving a set of data to be forgotten, the provider applies an unlearning method to update the model, which is then expected to avoid generating content that reflects the removed information.

However, despite the advantages, there is an overlooked problem in offering such services:

*Is it safe to blindly trust and unlearn user-submitted data without concern?*

A malicious user may submit a disguised forgetting dataset in an unlearning request to intentionally degrade model utility—particularly when the unlearned model is later provided to normal users. In this paper, by proposing a **Stealthy Attack (SA)**, we demonstrate that fine-tuning-based LLM unlearning—the most prominent and widely adopted category—suffers from this critical vulnerability.

Stealthy Attack (SA) is motivated by two fundamental limitations in current unlearning mechanisms. **First**, existing methods struggle to constrain the scope of unlearning effectiveness, as highlighted in prior studies [15, 16]. For instance, after unlearning the book "*Watermelon on the Moon*", a model is asked: "*Who is the author of Watermelon on the Moon? And where is the Eiffel Tower?*" Although the second question is unrelated, the model fails to answer it—an unintended consequence of overgeneralization from unlearning process. Other studies attribute this issue to the fact that unlearning in LLMs does not truly erase the forgetting data [17, 18, 19], but rather memorizes it as a signal to simulate ignorance or hallucinate [20]. **Second**, fine-tuning-based unlearning methods inadvertently treat benign tokens—those unrelated to the forgetting knowledge—as unlearning signals. This occurs because such methods cannot distinguish between tokens that encode the forgetting knowledge (referred to as target tokens) and unrelated benign tokens, as illustrated in Section 3.2. These limitations open the door for malicious users to attack the unlearning process by increasing the model's reliance on common, benign tokens such as "*please*" and "*then*". When the unlearned model is later deployed to normal users—who naturally use these benign tokens—the model's performance degrades. Stealthy Attack exploits this vulnerability by intentionally increasing the influence of the unlearning loss on selected benign tokens (referred to as **benign triggers**) through subtle frequency manipulation. Experiments in Section 5 demonstrate the effectiveness of this attack, revealing a critical vulnerability in current unlearning techniques.

To mitigate the above vulnerability and enhance the robustness of unlearning, we propose **Scope-aware Unlearning (SU)**, a simple yet effective improvement to the fine-tuning-based unlearning framework. In addition to the standard forgetting and retaining losses, we introduce a **scope term** that constrains the effect of unlearning to the appropriate region within the target knowledge. This is achieved by constructing *scope samples* that combine forgetting and retaining prompts, training the model to ignore unlearning signals when they appear in normal contexts. Our method requires no additional processing—only the addition of a single term to the unlearning objective. Experimental results show that this approach significantly improves robustness against SA. Moreover, it is model-agnostic and compatible with existing fine-tuning-based unlearning methods, making it easy to integrate into current workflows.

## 2 Related works

**Machine Unlearning.** Machine Unlearning is first proposed in the domain of computer vision and classification models [21, 22, 23, 24, 14]. Recently, it has been extended into the area of LLMs to mitigate the risks associated with undesirable data [25, 10, 18, 26, 25, 27, 28, 29]. Fine-tuning-based unlearning is the most widely used category due to its straightforward motivation and promising performance [15, 17, 30, 31]. The existing fine-tuning objectives can be divided into two categories. The first is Gradient Ascent (GA) [32, 9] and its variants [10, 11, 12, 13, 33, 34, 35, 36, 37, 38]. These methods fine-tune the LLM with the reversed training loss on forgetting data to negate the training impact. The second is the preference-based methods [9, 39, 40, 41, 42, 35, 43]. These methods do not aim at removing the influence of forget data. Instead, they instruct the model to generate human-preferred response to forgetting data such as "I don't know".

**The mechanism of unlearning in LLM.** Although the goal of unlearning, especially GA-based methods, is to remove the influence of forgetting data, more research works reveal that LLM unlearning is not actually unlearning or forgetting [19, 44, 17, 31]. For instance, Deeb et al. [17] demonstrate that fine-tuning with unrelated data can recover the knowledge that is unlearned. Ren et al. [20] find that the mechanism of GA-based methods is similar to preference-based unlearning from a hidden representation perspective. GA-based methods cannot precisely negate the training, but only memorize the data that should behave like unknown, treating the forgetting data as an unlearning signal. Once the signal exist in the prompt regardless of whether containing the unlearned question or not, the model would behave like unknown. Other works [15, 16] also find that when mixing forgetting data and retaining data in the prompt, the model would be dominated by unlearning signals and cannot answer the questions about retaining data either.

## 3 The vulnerability of LLM unlearning

### 3.1 Problem statement

In this subsection, we begin by revisiting the definitions of the unlearning problem and fine-tuning-based unlearning, and then outline the settings used to simulate the real-world scenarios.

**Unlearning in LLM and fine-tuning based methods.** Given an LLM $f$ and a forgetting dataset $\mathcal{D}_{\mathrm{f}}$, the goal of LLM unlearning is to get a model $f_{\mathrm{u}}$ that behaves as if it was never trained on $\mathcal{D}_{\mathrm{f}}$ (e.g., output "I don't know" or hallucinate an incorrect answers). Besides, $f_{\mathrm{u}}$ should also retain the model utility, i.e. the general text generation capabilities.

Following [20], the objective of fine-tuning-based unlearning can be formulated as

$$\operatorname*{argmin}_{\theta} \mathbb{E}_{(x,y)\in\mathcal{D}_{\mathrm{f}}}\left[L_{\mathrm{f}}(y,x,\theta)\right] + \lambda\mathbb{E}_{(x,y)\in\mathcal{D}_{\mathrm{r}}}\left[L_{\mathrm{r}}(y,x,\theta)\right], \tag{1}$$

where $\mathcal{D}_{\mathrm{r}}$ is the retaining dataset to preserve the model utility, and $\theta$ represents the parameters to be updated. The notation $(x,y)$ denotes an input-label pair of $\mathcal{D}_{\mathrm{f}}$ and $\mathcal{D}_{\mathrm{r}}$. The terms $L_{\mathrm{f}}$ and $L_{\mathrm{r}}$ denote the forgetting and retaining loss functions respectively, with $\lambda$ balancing them. The loss $L_{\mathrm{f}}$ aims at reducing the generation probability of forgetting data, while $L_{\mathrm{r}}$ maintains the model utility. Typically, $L_{\mathrm{f}}$ can be the negative training loss (i.e., Gradient Ascent [32]) or its variants [10, 12], and $L_{\mathrm{r}}$ corresponds to the training loss on $\mathcal{D}_{\mathrm{r}}$ or a regularization term (e.g., the KL divergence between $f$ and $f_{\mathrm{u}}$ [45]).

Note that the definition of $(x,y)$ can vary depending on the data format. In question-answering (QA) settings, $x$ typically represents the question and $y$ the corresponding answer. For non-QA formats such as pretraining data, the $(x,y)$ pair is often formulated as next-token prediction (i.e., $y = (x_2, x_3, \ldots, x_T)$, where $T$ is the length of $x$), and the loss is computed autoregressively.

**Unlearning settings.** Although various benchmarks have been proposed to measure the performance of unlearning methods, gaps still persist between the real-world scenarios and the benchmark tasks. In particular, the benchmark tasks usually provide the specific knowledge and corpus to unlearn, such as the synthetic book-author QAs in TOFU [9] and the celebrity identities in RWKU [46], while how to get the corpus can be flexible and diverse based on the practical usage in real-world scenarios. In this work, we consider the following unlearning service setting:

- User capability: The user provides the particular corpus of forgetting data (i.e., $\mathcal{D}_{\mathrm{f}}$) to the model builder. The user has no control of the unlearning algorithm.

- Model builder capability: The model builder conducts the unlearning algorithm to update the model. The model builder is allowed to protect the unlearning by pre-processing (such as data filtering) or post-processing (such as fine-tuning with clean data).

### 3.2 A Stealthy Attack

As mentioned above, the unlearning service can be potentially exploited by malicious users. To demonstrate this vulnerability, we propose a Stealthy Attack (SA). SA only modifies the forget data in a stealthy manner before submitting to the unlearning service. Once the model builder unlearns and releases the updated model, the utility will be reduced if the benign tokens are presented in the input. The intuition and the details of the proposed attack are as follows.

**Intuition.** The design of SA is inspired by the following two aspects.

First, as mentioned in Section 1, existing works [20, 15, 16] have found that the unlearning signals may severely disturb the utility on normal prompts. Based on [20], when unlearning signals (such as target tokens) are mixed in the normal prompts, the embedding space would be dominated by the unlearning signals and the model will be unable to response to the normal prompts. For example, suppose that a model has unlearned the knowledge of a book "*Watermelon on the Moon*". We ask it a target question and a normal question together like: "*Who is the author of Watermelon on the Moon? And where is Eiffel Tower?*". Even though the knowledge of "*Eiffel Tower*" is retained by the model, the unlearning effectiveness will also overgeneralize to the whole prompt and the model becomes unable to answer "*Where is Eiffel Tower?*" (More analysis on the reason for this phenomenon can be found in Appendix A). Based on this phenomenon, if we can induce the model to treat benign tokens as the unlearning signals, then when normal users use these tokens, they will trigger unlearning behaviors, which will severely degrade the model performance.

Second, in order to effectively induce the model to treat benign tokens as the unlearning signals, we take a closer look at the design of the unlearning loss functions. We hypothesize that the unlearning methods are unable to effectively distinguish benign and target tokens, and will link all of them to the unlearning signals. This implies a possible attack strategy: by increasing the frequency of benign tokens in the forgetting data, one can amplify their influence, causing them to be misinterpreted as unlearning signals. In the following, we use Gradient Ascent (GA) to illustrate the hypothesis:

In the ideal unlearning, if the unlearning corpus is "*The author of Watermelon on the Moon was born in 1988*", the expected unlearning behavior is to hallucinate/reject to answer when prompt contains the target tokens "*Watermelon on the Moon*". If the benign tokens "*The author*" show in a normal prompt alone, the model should behave normally. [2] However, by taking a closer look at the loss in GA, we find that the unlearning algorithm indeed does not distinguish the target tokens from benign tokens. Specifically, the forgetting loss of GA on a sequence $x = (x_1, x_2, \ldots, x_T)$ in the non-QA format is

$$L_{\mathrm{f}}(x, \theta) = -L_{\mathrm{train}}(x, \theta) = \sum_{t=1}^{T} \log p\left(x_t \mid x_{<t}, \theta\right), \tag{2}$$

where $L_{\mathrm{train}}$ is the next token prediction loss of autoregressive generation models, and $x_{<t} = (x_1, x_2, \ldots, x_{t-1})$ is the prefix. Due to the design of autoregressive LLMs, the prediction of $x_t$ is conditioned on the encoded representations of the prefix $x_{<t}$, denoted as $e_{<t}$, which is encoded by the model parameterized by $\theta$. Accordingly, Eq. (2) can be rewritten as:

$$L_{\mathrm{f}}(x, \theta) = \sum_{t=1}^{T} \log p\left(x_t \mid x_{<t}, \theta\right) = \sum_{t=1}^{T} \log p\left(x_t \mid e_{<t}, \theta\right), \tag{3}$$

In each term $\log p\left(x_t \mid e_{<t}, \theta\right)$, the semantic information of both target and benign tokens is entangled within the embeddings $e_{<t}$. This means that the semantics of benign tokens cannot be precisely disentangled or removed. In fact, current unlearning algorithms are not even designed to do so. As a result, when the model is trained to treat $e_{<t}$ as an unlearning signal through $L_{\mathrm{f}}$, it inevitably incorporates benign tokens into the forgetting process. For the aforementioned example, when $x_{<t}$ is "*The author of*" and $x_t$ is "*Watermelon*", the prefix only contains benign tokens, which means that benign tokens are definitely used as unlearning signals. When $x_{<t}$ is "*The author of Watermelon on the Moon was*" and $x_t$ is "*born*", "*born*" is highly related with "*author*". The unlearning has no motivation to use only "*Watermelon on the Moon*" as the signal to suppress the generation of "*born*".

The above two aspects of existing fine-tuning-based unlearning suggest that it is possible to reduce the model performance by inducing the unlearning methods taking some benign and common words such as "please" and "then" to be the unlearning signals. When these tokens are used in normal questions and prompts by the normal users, the model will pretend that they have no knowledge on these questions, leading to performance degradation.

*Remarks.* Although retaining loss is often used in unlearning, such as Negative Preference Optimization (NPO) [12], to recover the utility on normal prompts, it cannot precisely counteract the

---

[2] There are some tokens, like "*1988*", might be ambiguous on whether they are target tokens and benign tokens. Here we only discuss the clearly benign and target tokens which will not influence the conclusion.

Table 1: Template sets for TOFU and RWKU

| Dataset | Template set |
|---------|--------------|
| TOFU | "***Please** tell me* {original question}", "*Can you **please** explain* {original question}", "*Could you **please** tell me* {original question}", "{original question}, ***please**?*" |
| RWKU | "***Then*** {original sentence}" |

influence on benign tokens since the retaining data is not guaranteed to have the exact benign tokens as forgetting data.

**Stealthy Attack.** To enhance the impact on benign tokens and induce the model to treat them as unlearning signals, the Stealthy Attack increases their frequency in the forgetting data. We first define a corpus transformation function $\mathcal{T}$ as follows:

$$\mathcal{T}(x) = \begin{cases} \text{TEMPLATE}(x), & \text{if } r < \text{p,} \\ x, & \text{otherwise,} \end{cases} \quad \text{where } r \sim \mathcal{U}(0, 1). \quad (4)$$

Here, TEMPLATE$(x)$ rewrites the original sentence into a natural-sounding template that contains selected benign tokens (benign triggers) intended to serve as unlearning signals. The transformation probability p controls how often the template is applied. We design two sets of templates—one for QA corpora and another for non-QA corpora—using "please" and "then" as the target benign tokens (see Table 1). The template set is flexible and can be defined by the malicious users. Applying $\mathcal{T}$ to each sentence in $\mathcal{D}_f$ yields the modified dataset $\mathcal{D}_f'$, where benign token frequency is stealthily increased. Although the method is straightforward, it is both effective and stealthy, as evidenced by our experimental results in Section 5. The templates are short and natural (Table 1), making the modifications difficult to detect. In Section 5.4, these subtle modifications evade anomaly detection while successfully amplifying the unlearning signal.

## 4 Improving the robustness by Scope-aware Unlearning

To enhance the robustness of unlearning and address its vulnerability, a straightforward idea is to remove benign tokens from the unlearning signals. In this way, when other users include benign tokens in their prompts, the model's behavior would remain unaffected. However, this is challenging in practice because benign tokens are often common words or natural conjunctions and prepositions, making it difficult to exclude them from $e_{<t}$ and the unlearning loss. Moreover, the boundary between benign and target tokens is often ambiguous (such as "*born in 1988*" in the above case). Instead of relying on token exclusion, we draw inspiration from the intuition of SA—the unclear scope of unlearning effect—and propose a more principled solution: enforcing precision in the effective scope of unlearning. Specifically, we constrain unlearning signals to be effective only when answering questions related to the target knowledge. Once the forgetting effect is localized within the correct scope, we no longer need to worry about unintended disruptions to normal prompts.

**The overlooked issue in fine-tuning-based unlearning objectives.** In Eq. (1), the forgetting loss encourages the model to behave as if it is unaware when unlearning signals are present, while the retaining loss encourages normal behavior when such signals are absent. However, the model may interpret this simplistically as: whenever unlearning signals appear, it should behave as if unaware. In other words, the model may struggle to determine whether unlearning signals should be applied in normal prompts, leading to unintended utility degradation.

**Scope-aware Unlearning (SU).** To clarify and constrain the scope of unlearning, we introduce a *scope term* into the training objective using *scope samples*—prompts that embed unlearning signals within otherwise normal inputs. The model is fine-tuned to ignore the unlearning signals when responding to the normal parts of such prompts. Rather than constructing a separate dataset, we generate scope samples dynamically within each training batch. Specifically, we concatenate a prompt $x_f$ from forgetting set $\mathcal{D}_f$ and a prompt $x_r$ from retaining set $\mathcal{D}_r$ in the current batch, forming a scope sample denoted as $x_t \oplus x$. We then incorporate the scope term into the unlearning objective, resulting in the following formulation:

$$\underset{\boldsymbol{\theta}}{\arg\min} \, L_u + \eta \, \mathbb{E}_{(x,y) \in \mathcal{D}_r, x_t \in \mathcal{D}_f} \left[ L_{\text{train}}(y | x_t \oplus x, \boldsymbol{\theta}) \right], \quad (5)$$

Here, $L_u$ represents the original unlearning objective (e.g., GA or NPO), while the second term—the scope term—encourages the model to preserve its response ability to the normal portion of the input, even when unlearning signals are present. The coefficient $\eta$ controls the relative strength of this constraint.

In summary, our method does not require identifying or filtering benign triggers. Regardless of whether the unlearning signals originate from target tokens or benign tokens, the model is trained to restrict their effect to the intended scope, ensuring that normal prompts remain unaffected. The approach is simple and lightweight, introducing no additional data processing or operations—only an extra term in the unlearning objective.

## 5 Experiments

In this section, we first present the attack results of TOFU on vanilla unlearning methods across different models in Section 5.2. Then we show the results of improved robustness by SU in Section 5.3. In Section 5.4, we compare SU with two methods which are adapted from defenses against data poisoning attacks. Finally, we conduct the ablation studies in Section 5.5. Due to page limitation, we present the results of RWKU in Appendix E.4, the attack effectiveness on the inference-time unlearning method (Task Vector) in Appendix E.1, results across additional LLMs (Phi [47] and Zephyr [48]) in Appendix E.2, and an ablation study on the coefficient $\eta$ of SU in Appendix E.3.

### 5.1 Experimental settings

**Models, datasets, and unlearning methods** We use LLaMA 3.1 (8B) [49] and Mistral v0.3 (7B) [50] for our experiments on two datasets: TOFU (QA-format) and RWKU (non-QA-format). Following [9], we evaluate three unlearning methods on TOFU: Gradient Difference (GD) [11], NPO [12], and IDK [20], using both vanilla and LoRA fine-tuning. GD and NPO are GA-based methods, while IDK is a preference-based approach that trains the model to respond with statements like "I don't know." For RWKU [46], we apply GD and NPO, which are compatible with non-QA corpora.

**Implementation details.** We set the transformation probability p in Eq. (4) to 0.33. The final token insertion rates are 2.13% for TOFU and 1.12% for RWKU. For TOFU, we follow the implementation of [13]. For RWKU, our implementation builds on [13] and [46], with fine-tuning details in Appendix C. LoRA is used with a rank of 8. All experiments are conducted on H100 GPUs. All other details are in Appendix C.

**Metrics.** We focus on ROUGE-L Recall as the main metric. It is representative and shared by the original settings in TOFU [9] and RWKU [46]. Other metrics of TOFU are presented in Appendix E.6 to save page space. ROUGE-L Recall assesses correctness of the output text by measuring how many groundtruth tokens are generated. Lower scores on forgetting data indicate better erasing, while higher scores on normal data indicates better utility.

### 5.2 Attack results of Stealthy Attack

In this subsection, we present the attack results of SA on TOFU. We evaluate each method across three dimensions: (1) how well the target data is forgotten (*unlearning effectiveness*), (2) model utility on clean inputs (*clean utility*), and (3) model utility with benign triggers (*benign-trigger utility*). The results are averaged on 5 random seeds and the standard deviation is presented in Appendix E.5. The results on RWKU hold consistent conclusion and are presented in Appendix E.4.

TOFU is a synthetic dataset containing fake books and authors. LLMs are first fine-tuned on the full dataset, after which a subset is designated as forgetting data for unlearning, while the remaining data serves as retaining data for utility. TOFU also includes two sets of real authors and real-world knowledge for utility testing. As shown in Table 2, SA significantly reduces utility when benign-trigger tokens are present in the test set. This vulnerability is consistent across models and unlearning methods. A detailed analysis of each unlearning method is provided below.

• For *GD*, SA reduces the benign-trigger utility of vanilla GD by approximately 60%–86%, highlighting the severity of the unlearning vulnerability. For example, when unlearning 10% of the TOFU data, the benign-trigger utility drops from 0.683 to 0.160 on LLaMA and from 0.405 to 0.056 on Mistral—rendering the model nearly unusable when benign triggers are present. Notably, the

Table 2: Attack Results of TOFU. $p_{tgt}$ represents the proportion of forgetting data within the entire synthetic dataset. "No unlearn" is the results before unlearning. The column of "Unlearn" refers to the unlearning effectiveness.

| | $L_u$ | $p_{tgt}$ | Attack | Unlearn↓ | Clean utility↑ Average (Retain/Fact/World) | Benign-trigger utility↑ Average (Retain/Fact/World) |
|---|---|---|---|---|---|---|
| **LLaMA** | No unlearn | 5% 10% | N/A | 0.991 0.992 | 0.940 (0.992/0.939/0.888) | 0.926 (0.977/0.919/0.882) |
| | GD | 5% | no | 0.005 | 0.706 (0.496/0.854/0.766) | **0.662** (0.488/0.803/0.694) |
| | | | SA | **0.002** | **0.708** (0.525/0.762/0.835) | 0.236 (0.178/0.244/0.287) |
| | | 10% | no | **0.005** | 0.702 (0.483/0.838/0.785) | **0.683** (0.480/0.853/0.715) |
| | | | SA | 0.008 | **0.752** (0.524/0.869/0.862) | 0.160 (0.187/0.156/0.136) |
| | NPO | 5% | no | **0.203** | 0.747 (0.611/0.746/0.883) | **0.741** (0.592/0.763/0.868) |
| | | | SA | 0.247 | **0.777** (0.592/0.860/0.878) | 0.552 (0.402/0.609/0.646) |
| | | 10% | no | **0.199** | 0.734 (0.549/0.801/0.851) | **0.732** (0.541/0.796/0.860) |
| | | | SA | 0.225 | **0.765** (0.556/0.851/0.888) | 0.217 (0.188/0.197/0.267) |
| | IDK | 5% | no | **0.023** | **0.664** (0.568/0.617/0.806) | **0.649** (0.567/0.585/0.796) |
| | | | SA | 0.025 | 0.654 (0.593/0.565/0.803) | 0.354 (0.449/0.192/0.421) |
| | | 10% | no | **0.023** | 0.549 (0.575/0.363/0.710) | **0.496** (0.552/0.343/0.594) |
| | | | SA | 0.029 | **0.574** (0.578/0.388/0.755) | 0.164 (0.313/0.052/0.128) |
| **Mistral** | No unlearn | 5% 10% | N/A | 0.999 0.998 | 0.680 (0.999/0.358/0.683) | 0.658 (0.993/0.363/0.618) |
| | GD | 5% | no | **0.001** | **0.472** (0.697/0.259/0.460) | **0.462** (0.690/0.224/0.471) |
| | | | SA | 0.021 | 0.443 (0.719/0.206/0.403) | 0.183 (0.369/0.082/0.099) |
| | | 10% | no | 0.000 | 0.426 (0.822/0.212/0.245) | **0.405** (0.804/0.168/0.242) |
| | | | SA | **0.002** | **0.378** (0.668/0.148/0.318) | 0.056 (0.141/0.010/0.018) |
| | NPO | 5% | no | 0.186 | **0.568** (0.843/0.305/0.557) | **0.551** (0.824/0.288/0.541) |
| | | | SA | **0.153** | 0.566 (0.805/0.314/0.578) | 0.243 (0.360/0.134/0.235) |
| | | 10% | no | **0.027** | 0.573 (0.875/0.209/0.636) | **0.556** (0.872/0.195/0.602) |
| | | | SA | 0.068 | **0.584** (0.857/0.278/0.617) | 0.012 (0.026/0.004/0.005) |
| | IDK | 5% | no | **0.023** | 0.511 (0.825/0.208/0.500) | **0.469** (0.797/0.162/0.448) |
| | | | SA | 0.024 | **0.540** (0.841/0.196/0.582) | 0.193 (0.425/0.054/0.100) |
| | | 10% | no | **0.025** | 0.499 (0.825/0.183/0.488) | **0.437** (0.806/0.133/0.373) |
| | | | SA | 0.027 | **0.515** (0.851/0.171/0.524) | 0.192 (0.463/0.040/0.074) |

clean utility remains largely unaffected, especially on LLaMA, indicating that SA can precisely use designate benign triggers as unlearning signals.

● For *NPO*, we observe similar results to GD. SA still significantly degrades benign-trigger utility, revealing a clear vulnerability in vanilla NPO. Notably, the attack is more pronounced when unlearning 10% of TOFU data compared to 5%. For example, on LLaMA, the benign-trigger utility drops by 0.189 with 5% data forgetting, but by 0.524 with 10% data forgetting. Since unlearning 10% of the data requires more steps in our setting (following Fan et al. [13], where each epoch contains more data), this suggests that additional unlearning steps may amplify the impact on benign tokens.

● For *IDK*, SA reduces the benign-trigger utility by 45%–67% on LLaMA and 56%–59% on Mistral. This verifies that the preference-based methods also cannot distinguish benign tokens from unlearning signals and SA can reduce the utility successfully.

In summary, SA exposes a critical vulnerability in fine-tuning-based unlearning methods across all paradigms. Despite differences in unlearning strategies—whether GD, NPO, or preference-driven IDK—none are robust against the stealthy misuse of benign tokens as unlearning signals. The attack consistently degrades benign-trigger utility while leaving clean utility largely intact, making it difficult to detect. These results underscore the urgent need for the improvement on the robustness of unlearning methods.

Table 3: Robustness Results of TOFU. The columns of "Unlearn", "Clean-util", and "Trigger-util" refer to the unlearning effectiveness, clean utility and benign-trigger utility respectively. All the results of utility is averaged over threes subsets of retaining data, real authors and world facts.

| $L_u$ | $p_{tgt}$ | Method | LLaMA | | | Mistral | | |
|---|---|---|---|---|---|---|---|---|
| | | | Unlearn↓ | Clean-util↑ | Trigger-util↑ | Unlearn↓ | Clean-util↑ | Trigger-util↑ |
| GD | 5% | Vanilla | **0.002** | **0.708** | 0.236 | 0.021 | **0.443** | 0.183 |
| | | SU | 0.020 | 0.655 | **0.558** | **0.016** | 0.411 | **0.307** |
| | 10% | Vanilla | **0.008** | **0.752** | 0.160 | **0.002** | 0.378 | 0.056 |
| | | SU | 0.199 | 0.750 | **0.684** | 0.005 | **0.410** | **0.216** |
| NPO | 5% | Vanilla | 0.247 | **0.777** | 0.552 | **0.153** | **0.566** | 0.243 |
| | | SU | **0.203** | 0.699 | **0.672** | 0.255 | 0.412 | **0.395** |
| | 10% | Vanilla | **0.225** | **0.765** | 0.217 | 0.068 | **0.584** | 0.012 |
| | | SU | 0.256 | 0.763 | **0.734** | **0.007** | 0.438 | **0.198** |
| IDK | 5% | Vanilla | **0.025** | 0.654 | 0.354 | **0.024** | **0.540** | 0.193 |
| | | SU | 0.026 | **0.660** | **0.608** | **0.024** | 0.468 | **0.375** |
| | 10% | Vanilla | **0.029** | 0.574 | 0.164 | **0.027** | **0.515** | 0.192 |
| | | SU | **0.029** | **0.689** | **0.596** | 0.029 | 0.455 | **0.373** |

Table 4: Comparison with defenses for data poisoning attack. ROUGE-L Recall averaged over GD, NPO, and IDK.

| | Method | Unlearn↓ | Clean utility↑ | Benign-trigger utility↑ |
|---|---|---|---|---|
| LLaMA | Continuous fine-tuning | 0.343 (0.358/0.310/0.362) | **0.776** (0.785/0.751/0.791) | 0.537 (0.658/0.221/0.731) |
| | Anomaly detection | **0.124** (0.006/0.312/0.053) | 0.676 (0.677/0.754/0.596) | 0.336 (0.308/0.418/0.282) |
| | SU | 0.161 (0.199/0.256/0.029) | 0.734 (0.750/0.763/0.689) | **0.671** (0.684/0.734/0.596) |
| Mistral | Continuous fine-tuning | 0.304 (0.323/0.368/0.220) | **0.553** (0.528/0.566/0.565) | 0.213 (0.192/0.002/0.444) |
| | Anomaly detection | 0.072 (0.003/0.136/0.078) | 0.508 (0.476/0.524/0.524) | 0.181 (0.224/0.118/0.200) |
| | SU | **0.014** (0.005/0.007/0.029) | 0.434 (0.410/0.438/0.455) | **0.262** (0.216/0.198/0.373) |

## 5.3 Robustness results of Scope-aware Unlearning

In Table 3, we present the robustness improvements achieved by SU under SA compared to vanilla unlearning methods under SA. We can see that SU significantly improves benign-trigger utility while preserving unlearning effectiveness. A detailed analysis for each unlearning method is provided below.

• For *GD*, SU significantly improves robustness under SA. For example, on LLaMA, when removing 10% of TOFU data, for vanilla unlearning, SA reduces the benign-trigger utility to 0.160, while SU recovers the utility to 0.684. On Mistral, where SA is more effective, SU still improves the benign-trigger utility by 0.125 with 5% forgetting, and 0.160 with 10% forgetting.

• For *NPO*, SU shows strong resilience to SA, especially on LLaMA. On LLaMA, it effectively recovers the benign-trigger utility to a level nearly identical to that of the no-attack scenario, indicating that SU almost fully neutralizes the SA's impact.

• For *IDK*, SU consistently recovers benign-trigger utility across all settings. On LLaMA, while vanilla IDK drops to 0.354 and 0.164 in benign-trigger utility under SA, SU recovers it to 0.608 and 0.596. On Mistral, SU improves benign-trigger utility from 0.19 to 0.37 which is only around 0.05 to fully neutralizing the SA's impact.

In summary, SU consistently improves the robustness of all unlearning methods against SA by substantially restoring benign-trigger utility while preserving clean utility and unlearning effectiveness, demonstrating its effectiveness in mitigating unintended vulnerability of unlearning.

## 5.4 Existing defenses against LLM data-poisoning attack are ineffective in unlearning

In this subsection, we compare SU with two methods adapted from defenses against data poisoning in LLMs: continuous fine-tuning [51, 52] and abnormally detection [53, 54]. However, these

methods are not well-suited for unlearning settings. We first discuss their limitations in the context of unlearning, and then present experimental results that illustrate their ineffectiveness.

(1) **Continuous fine-tuning.** Due to the catastrophic forgetting phenomenon in neural networks [55, 56], it is possible that the memorization of benign triggers is erased through continuous fine-tuning with clean data after unlearning. However, the intended unlearning effect may also be eliminated [17], undermining the purpose of unlearning. Additionally, continuous fine-tuning incurs significant computational costs, making the approach economically impractical. In Table 5.4, we evaluate this method by further fine-tuning models—previously unlearned on TOFU data—using an auxiliary synthetic dataset composed of fake books and authors. The results indicate that the unlearning effectiveness of continuous fine-tuning is substantially worse than that of Scope-aware Unlearning (SU). Specifically, the unlearning effectiveness of LLaMA and Mistral both increased to higher than 0.3. On Mistral, the catastrophic forgetting on forgetting data (which is 0.304) is even more severe than the catastrophic forgetting benign trigger (which is 0.213).

(2) **Anomaly detection.** We use a straightforward method of high loss removal for anomaly detection [54]. As discussed in Section 3.2, the templates used in SA are designed to be natural and subtle, resulting in minimal distributional shifts. Consequently, these modifications are less likely to be detected by standard anomaly detection methods. In Table 4, we remove the top high-loss samples at a ratio of $p$. The results show that this approach fails to prevent SA from degrading the benign-trigger utility. The benign-trigger utility still reduces to 0.336 on LLaMA and 0.181 on Mistral.

## 5.5 Ablation studies

In this subsection, we further (1) evaluate the generalizability SU under parameter-efficient fine-tuning settings and (2) investigate how model utility evolves during the unlearning process.

(1) In Table 5, we present the results of removing 10% of the TOFU data using LoRA, to validate our findings across different fine-tuning frameworks. Although LoRA freezes most of the model parameters and updates only a small subset, it remains highly susceptible to benign triggers. The ROUGE-L recall on benign-trigger utilities is even lower than that on the forgetting data under GD and NPO, indicating that benign tokens are heavily treated as unlearning signals. Scope-aware Unlearning (SU) effectively addresses this issue, restoring utility across all metrics. These results confirm that SU is compatible with parameter-efficient fine-tuning approaches such as LoRA.

Table 5: Unlearning results using LoRA

| $L_u$ | Method | Attack | Unlearn↓ | Utility↑ | Trigger↑ |
|---|---|---|---|---|---|
| GD | Vanilla | no | 0.478 | **0.728** | **0.737** |
|  | Vanilla | SA | 0.354 | 0.546 | 0.093 |
|  | SU | SA | **0.220** | 0.608 | 0.458 |
| NPO | Vanilla | no | 0.426 | 0.743 | 0.723 |
|  | Vanilla | SA | **0.401** | 0.715 | 0.345 |
|  | SU | SA | 0.473 | **0.767** | **0.763** |
| IDK | Vanilla | no | **0.023** | 0.814 | 0.655 |
|  | Vanilla | SA | 0.057 | **0.825** | 0.379 |
|  | SU | SA | 0.042 | 0.799 | **0.685** |

(2) To investigate how model utility evolves during the unlearning process, Figure 1 presents the unlearning effectiveness and benign-trigger utility on the TOFU dataset under SA. We observe that vanilla methods begin to treat both the forgetting data and benign tokens as unlearning signals at approximately the same training step (between steps 10 and 20), suggesting that the model fails to effectively distinguish between target and benign tokens. In contrast, SU preserves high benign-trigger utility throughout the process, demonstrating its ability to constrain the unlearning effect to the intended scope.

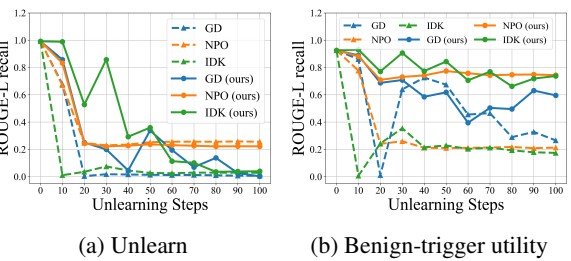

(a) Unlearn      (b) Benign-trigger utility

Figure 1: Unlearning in different fine-tuning steps

# 6   Conclusions

We identify a critical vulnerability in fine-tuning-based LLM unlearning, where benign tokens can be misused as unlearning signals to degrade model utility. To address this, we propose Scope-aware Unlearning, which constrains unlearning effects to the correct context without requiring token-level filtering. SU is simple, compatible with existing methods, and significantly improves robustness against stealthy attacks across models and datasets, offering a practical step toward safer unlearning.

**Limitations and broader impacts.** One limitation of this work is that it focuses on fine-tuning-based unlearning methods; the vulnerability of other unlearning approaches remains an open question. We leave this exploration for future work. Our contributions include a red-teaming attack and a robustness enhancement method, which we hope will motivate further research on improving the reliability of unlearning. We foresee no negative societal impacts.

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

## A Explanations of unclear scope

The unclear scope of unlearning effectiveness is also a key factor behind the vulnerability. Although the forgetting loss in Eq. (1) on forgetting data can train the LLM to memorize and suppress the forgetting data, the unlearning is not fine-tuned to learn the correct scope to use its effectiveness. If we look at the two terms in Eq. (1), forgetting loss teaches the model to behave like unaware when there is unlearning signals, while retaining loss teaches the model to behave normally when there is no unlearning signal. However, there is no intention in Eq. (1) to tell the model that unlearning signals should only be effective for target knowledge. The model may be uncertain whether the unlearning signals should be used for normal prompts. The empirical results in existing works [20, 15, 16] have shown that once there is unlearning signals in normal prompts, the embedding space would be dominated by the unlearning signals and model will be unable to response to the normal prompts. For example, we ask target question and normal question together like: "*Where is Eiffel Tower? And who is the author of Watermelon on the Moon?*" If the model has unlearned "Watermelon on the Moon", its unlearning effectiveness would also overgeneralize to the whole prompt and the model becomes unable to answer "Where is Eiffel Tower?", too.

## B Details of unlearning methods used in this paper

Following Ren et al. [20], we use three unlearning methods in this paper, GD, NPO and IDK.

GD applies the negative standard next-token prediction loss as $L_f$, and use the standard next-token prediction loss as $L_r$.

NPO constrains the divergence from the initial checkpoint to regulate strength of GA. GA and GD is aggressive because of the unlimited negative loss, while NPO is less aggresive and prevent the fast reduction of utility of GA. In this paper, we also use the standard next-token prediction loss as $L_r$.

IDK use the standard next-token prediction loss as both $L_f$ and $L_r$. But it use responses like "I don't know" for $L_f$.

## C Implementation details

For TOFU, all the implementations are based on the code of [13] with default parameters. For RWKU, we also use the code of [13], with a set of hyper-parameters as shown in Table 6. In additional to the training parameters in vanilla unlearning methods, we also include the values of new hyper-parameter $\eta$, which controls the strength in Table 7. We set epoch as 10 for all the experiments following [13].

Table 6: Hyper parameters

| Dataset | Model | $L_u$ | Hyper-parameter | value |
|---------|-------|-------|-----------------|-------|
| TOFU | LLaMA | GD | LR | 1e-5 |
| | | NPO | LR | 1e-5 |
| | | IDK | LR | 1e-5 |
| | Mistral | GD | LR | 1e-5 |
| | | NPO | LR | 1e-5 |
| | | IDK | LR | 1e-5 |
| RWKU | LLaMA | GD | LR | 1.8e-6 |
| | | NPO | LR | 1.4e-5 |
| | Mistral | GD | LR | 6e-7 |
| | | NPO | LR | 6e-6 |

## D Similar works on traditional classification models

Two works study similar problems [57, 58], but they do not focus on LLMs and require strong assumptions of model access to perform the attacks. Moreover, they do not provide any effective defense method.

Table 7: Hyper parameters

| Dataset | $L_\mathrm{u}$ | Model | Hyper-parameter | value |
|---|---|---|---|---|
| TOFU | LLaMA | GD (SU) | $\eta$ | 2 (for 5% removal) 12 (for 15% removal) |
| | | NPO (SU) | $\eta$ | 5 |
| | | IDK (SU) | $\eta$ | 5 |
| | Mistral | GD (SU) | $\eta$ | 5 |
| | | NPO (SU) | $\eta$ | 5 |
| | | IDK (SU) | $\eta$ | 5 |
| RWKU | LLaMA | GD (SU) | $\eta$ | 10 |
| | | NPO (SU) | $\eta$ | 40 |
| | Mistral | GD (SU) | $\eta$ | 10 |
| | | NPO (SU) | $\eta$ | 40 |

Table 8: Attack performance on Task Vector.

| $p_\mathrm{tgt}$ | Method | Unlearn↓ | LLaMA Clean-util↑ | Trigger-util↑ |
|---|---|---|---|---|
| 5% | SA | 0.362 | 0.730 | 0.615 |
| 10% | SA | 0.380 | 0.735 | 0.640 |

# E   Additional experiment results

## E.1   Attack performance on inference-time unlearning method

We conducted additional experiments on Task Vector, as shown in Table 8. The results show that our attack is also effective against Task Vector. Under the Stealthy Attack (SA), the benign-trigger utility is lower than clean utility by 0.115 (when removing 5%) and 0.095 (when removing 10%). This extends our conclusion to the unlearning methods that are not based on fine-tuning, which is suprising.

## E.2   Experiments on additional types of LLMs

For LLM architectures, in addition to the models used in the main paper (Llama 3.1 and Mistral), we include two additional types of LLMs: Phi [47] and Zephyr [48] in Table 9. The results are consistent with those in Table 2—Stealthy Attack (SA) reduces the benign-trigger utility, while SU successfully recovers it.

## E.3   Ablation study on the coefficient $\eta$

We conducted the ablation study on the coefficient $\eta$, the coefficient of Scope-aware term in SU in Table 10

On GD, as $\eta$ increases, the benign-trigger utility initially improves. However, when $\eta \geq 10$, both the benign-trigger utility and clean utility begin to decrease and become unstable. We conjecture that this

Table 9: Unlearning results on additional types of LLMs.

| $L_\mathrm{u}$ | LLM | Method | Attack | Unlearn↓ | Utility↑ | Trigger↑ |
|---|---|---|---|---|---|---|
| NPO | Phi | Vanilla | no | 0.388 | 0.618 | 0.550 |
| | | Vanilla | SA | 0.450 | 0.559 | 0.428 |
| | | SU | SA | 0.384 | 0.598 | 0.564 |
| | Zephyr | Vanilla | no | 0.130 | 0.580 | 0.550 |
| | | Vanilla | SA | 0.114 | 0.410 | 0.352 |
| | | SU | SA | 0.165 | 0.595 | 0.491 |

Table 10: Unlearning results on different values of $\eta$.

| $L_{\mathrm{u}}$ | $\eta$ | Unlearn↓ | Utility↑ | Trigger↑ |
|---|---|---|---|---|
| GD | 1 | 0.213 | 0.677 | 0.600 |
| | 2 | 0.019 | 0.699 | 0.644 |
| | 5 | 0.019 | 0.656 | 0.639 |
| | 10 | 0.022 | 0.423 | 0.436 |
| | 20 | 0.148 | 0.488 | 0.507 |
| NPO | 2 | 0.203 | 0.722 | 0.683 |
| | 5 | 0.193 | 0.718 | 0.672 |
| | 10 | 0.223 | 0.710 | 0.698 |
| | 50 | 0.250 | 0.727 | 0.714 |
| IDK | 2 | 0.022 | 0.651 | 0.615 |
| | 5 | 0.025 | 0.655 | 0.606 |
| | 10 | 0.023 | 0.663 | 0.615 |
| | 50 | 0.022 | 0.652 | 0.606 |

Table 11: Results of RWKU. "Unlearn" refers to the unlearning effectiveness, and "Clean" refers to the model before unlearning.

| LLM | $L_{\mathrm{u}}$ | Method | Method | Unlearn↓ Average (FB/QA/AA) | Clean utility↑ Average (FB/QA/AA) | Benign-trigger utility↑ Average (FB/QA/AA) |
|---|---|---|---|---|---|---|
| LLaMA | | No unlearn | N/A | 0.785 (0.755/0.804/0.796) | 0.807 (0.774/0.820/0.828) | 0.775 (0.733/0.808/0.785) |
| | GD | Vanilla | no | 0.009 (0.010/0.017/0.000) | 0.599 (0.613/0.615/0.570) | 0.607 (0.621/0.649/0.551) |
| | | Vanilla | SU | 0.016 (0.010/0.025/0.013) | 0.555 (0.536/0.581/0.549) | 0.415 (0.380/0.452/0.411) |
| | | SU | SU | 0.009 (0.010/0.017/0.000) | 0.606 (0.715/0.589/0.513) | 0.586 (0.657/0.598/0.503) |
| | NPO | Vanilla | no | 0.081 (0.099/0.076/0.067) | 0.508 (0.508/0.468/0.547) | 0.450 (0.493/0.360/0.496) |
| | | Vanilla | SU | 0.072 (0.055/0.073/0.087) | 0.521 (0.513/0.488/0.562) | 0.382 (0.399/0.337/0.410) |
| | | SU | SU | 0.071 (0.039/0.096/0.078) | 0.547 (0.549/0.491/0.601) | 0.510 (0.503/0.477/0.551) |
| Mistral | | No unlearn | N/A | 0.834 (0.887/0.802/0.812) | 0.842 (0.814/0.865/0.847) | 0.812 (0.819/0.806/0.810) |
| | GD | Vanilla | no | 0.009 (0.010/0.017/0.000) | 0.610 (0.697/0.569/0.563) | 0.589 (0.676/0.551/0.542) |
| | | Vanilla | SU | 0.009 (0.010/0.017/0.000) | 0.559 (0.648/0.560/0.468) | 0.289 (0.298/0.287/0.284) |
| | | SU | SU | 0.022 (0.021/0.000/0.047) | 0.835 (0.857/0.853/0.794) | 0.797 (0.795/0.818/0.779) |
| | NPO | Vanilla | no | 0.053 (0.053/0.050/0.055) | 0.583 (0.657/0.518/0.574) | 0.536 (0.614/0.492/0.502) |
| | | Vanilla | SU | 0.064 (0.054/0.081/0.059) | 0.538 (0.584/0.493/0.536) | 0.392 (0.454/0.389/0.333) |
| | | SU | SU | 0.151 (0.102/0.232/0.120) | 0.618 (0.685/0.541/0.626) | 0.506 (0.599/0.448/0.470) |

is due to the divergent nature of GD's objective (as discussed in [2]), which makes the fine-tuning process unstable.

On NPO, the performance is more stable. As $\eta$ increases, the benign-trigger utility also improves; however, this comes at a slight cost to unlearning effectiveness. For instance, when $\eta = 50$, the benign-trigger utility increases to 0.714, but the unlearning effectiveness worsens by 0.05 compared to $\eta = 2$.

On IDK, $\eta$ has very little impact on the utility. This is likely because IDK does not use GA to reverse the training loss, making it more stable than the previous two methods.

### E.4 RWKU

**RWKU.** Table 11 presents the results of unlearning real celebrity identities from the RWKU dataset. The corpus is in a non-QA format and does not require prior fine-tuning for the model to learn from it. In each experiment, we unlearn the identity of one celebrity, use a second celebrity's corpus as retaining data, and evaluate utility on a third celebrity. The evaluation of utility covers three formats: fill-in-the-blank (FB), question answering (QA), and adversarially crafted questions (AA). All results are averaged over five sets of different celebrities in independent runs.

On LLaMA, SA reduces benign-trigger utility by 31% under GD and 15% under NPO. On Mistral, the degradation is even more pronounced: 51% under GD and 27% under NPO. The attack appears

Table 12: STD of attack results of TOFU.

| | $L_\mathrm{u}$ | $p_\mathrm{tgt}$ | Method | Unlearn | Clean utility Average (Retain/Fact/World) | Benign-trigger utility Average (Retain/Fact/World) |
|---|---|---|---|---|---|---|
| LLaMA | GD | 5% | Vanilla | 0.002 | 0.011 (0.036/0.052/0.031) | 0.270 (0.183/0.286/0.343) |
| | | | SU | 0.001 | 0.067 (0.043/0.124/0.039) | 0.126 (0.048/0.167/0.183) |
| | | 10% | Vanilla | 0.006 | 0.010 (0.020/0.024/0.012) | 0.188 (0.145/0.234/0.187) |
| | | | SU | 0.109 | 0.018 (0.024/0.030/0.016) | 0.129 (0.022/0.176/0.200) |
| | NPO | 5% | Vanilla | 0.066 | 0.005 (0.013/0.010/0.021) | 0.056 (0.047/0.097/0.100) |
| | | | SU | 0.010 | 0.012 (0.010/0.031/0.013) | 0.008 (0.011/0.032/0.009) |
| | | 10% | Vanilla | 0.033 | 0.012 (0.017/0.022/0.012) | 0.198 (0.169/0.182/0.247) |
| | | | SU | 0.007 | 0.009 (0.013/0.030/0.007) | 0.015 (0.014/0.029/0.020) |
| | IDK | 5% | Vanilla | 0.003 | 0.013 (0.013/0.034/0.013) | 0.115 (0.098/0.098/0.158) |
| | | | SU | 0.003 | 0.014 (0.014/0.031/0.014) | 0.034 (0.012/0.065/0.028) |
| | | 10% | Vanilla | 0.001 | 0.011 (0.010/0.023/0.012) | 0.030 (0.055/0.016/0.028) |
| | | | SU | 0.003 | 0.013 (0.012/0.027/0.023) | 0.025 (0.010/0.059/0.019) |
| Mistral | GD | 5% | Vanilla | 0.021 | 0.077 (0.024/0.082/0.164) | 0.185 (0.323/0.097/0.144) |
| | | | SU | 0.009 | 0.044 (0.047/0.024/0.125) | 0.047 (0.020/0.024/0.129) |
| | | 10% | Vanilla | 0.002 | 0.106 (0.079/0.062/0.200) | 0.060 (0.140/0.021/0.040) |
| | | | SU | 0.005 | 0.059 (0.088/0.109/0.104) | 0.052 (0.093/0.053/0.060) |
| | NPO | 5% | Vanilla | 0.033 | 0.021 (0.013/0.026/0.048) | 0.154 (0.226/0.081/0.160) |
| | | | SU | 0.014 | 0.013 (0.019/0.016/0.028) | 0.021 (0.035/0.015/0.034) |
| | | 10% | Vanilla | 0.028 | 0.026 (0.028/0.060/0.018) | 0.018 (0.034/0.009/0.011) |
| | | | SU | 0.004 | 0.022 (0.021/0.037/0.047) | 0.132 (0.259/0.044/0.100) |
| | IDK | 5% | Vanilla | 0.003 | 0.010 (0.014/0.008/0.016) | 0.102 (0.181/0.038/0.101) |
| | | | SU | 0.002 | 0.004 (0.012/0.009/0.013) | 0.025 (0.018/0.010/0.062) |
| | | 10% | Vanilla | 0.004 | 0.005 (0.008/0.016/0.011) | 0.041 (0.078/0.010/0.050) |
| | | | SU | 0.002 | 0.020 (0.016/0.022/0.034) | 0.020 (0.018/0.010/0.042) |

more effective against GD, likely due to its more aggressive unlearning approach, which amplifies the influence of unlearning signals on benign tokens. In terms of robustness, SU successfully recovers the benign-trigger utility to nearly the same level as vanilla unlearning without attack on both models. Notably, the clean utility under SU is even higher than that of vanilla unlearning—a phenomenon not observed in TOFU. In contrast, TOFU shows minimal variation in clean utility across all settings (vanilla with/without SA and SU with SA). We hypothesize that this difference stems from the larger distributional gap between the forgetting and retaining sets in RWKU compared to TOFU. The scope term in SU reinforces correct responses from the retaining data (as used in unclear-scope samples), which may inadvertently enhance clean utility. In TOFU, this effect is diminished because the forgetting loss suppresses forgetting knowledge that closely resembles the retaining data, as both are drawn from the same synthetic distribution.

## E.5 Standard deviation

We report STD in Table 12.

## E.6 Other metrics in TOFU

Following the settings of TOFU, we also use metrics Probability and Truth Ratio.

**Probability Metric.** For a question-answer pair $(q, a)$, the conditional probability is computed as:

$$P(a \mid q)^{1/|a|}$$

where $|a|$ is the number of tokens in the answer $a$. This normalization accounts for the length of the answer.

Table 13: Probability of Attack Results of TOFU.

| | $L_u$ | $p_{tgt}$ | Attack | Unlearn | Clean utility Average (Retain/Fact/World) | Benign-trigger utility Average (Retain/Fact/World) |
|---|---|---|---|---|---|---|
| LLaMA | GD | 5% | no | 0.340 | 0.843 (0.986/0.774/0.769) | 0.833 (0.986/0.764/0.750) |
| | | | SA | 0.000 | 0.634 (0.687/0.630/0.585) | 0.319 (0.216/0.364/0.377) |
| | | 10% | no | 0.000 | 0.554 (0.654/0.496/0.511) | 0.521 (0.642/0.423/0.498) |
| | | | SA | 0.000 | 0.659 (0.710/0.635/0.632) | 0.291 (0.235/0.319/0.317) |
| | NPO | 5% | no | 0.016 | 0.645 (0.766/0.546/0.622) | 0.632 (0.757/0.525/0.614) |
| | | | SA | 0.020 | 0.627 (0.751/0.555/0.576) | 0.401 (0.372/0.404/0.428) |
| | | 10% | no | 0.025 | 0.599 (0.730/0.465/0.602) | 0.580 (0.723/0.454/0.563) |
| | | | SA | 0.052 | 0.594 (0.754/0.474/0.555) | 0.215 (0.064/0.284/0.298) |
| | IDK | 5% | no | 0.467 | 0.621 (0.870/0.476/0.517) | 0.603 (0.856/0.457/0.497) |
| | | | SA | 0.488 | 0.628 (0.874/0.484/0.527) | 0.594 (0.828/0.451/0.503) |
| | | 10% | no | 0.534 | 0.614 (0.872/0.459/0.512) | 0.594 (0.855/0.445/0.483) |
| | | | SA | 0.558 | 0.617 (0.873/0.458/0.520) | 0.561 (0.800/0.414/0.468) |
| Mistral | GD | 5% | no | 0.000 | 0.578 (0.838/0.436/0.461) | 0.556 (0.834/0.409/0.426) |
| | | | SA | 0.000 | 0.587 (0.858/0.426/0.478) | 0.380 (0.487/0.321/0.332) |
| | | 10% | no | 0.000 | 0.596 (0.895/0.417/0.477) | 0.580 (0.892/0.403/0.446) |
| | | | SA | 0.000 | 0.559 (0.821/0.398/0.456) | 0.242 (0.200/0.252/0.274) |
| | NPO | 5% | no | 0.018 | 0.573 (0.920/0.400/0.399) | 0.560 (0.915/0.379/0.385) |
| | | | SA | 0.020 | 0.561 (0.906/0.364/0.412) | 0.385 (0.501/0.315/0.338) |
| | | 10% | no | 0.001 | 0.578 (0.939/0.375/0.421) | 0.568 (0.936/0.368/0.401) |
| | | | SA | 0.012 | 0.569 (0.924/0.365/0.417) | 0.216 (0.041/0.304/0.303) |
| | IDK | 5% | no | 0.568 | 0.573 (0.970/0.353/0.395) | 0.566 (0.966/0.343/0.390) |
| | | | SA | 0.597 | 0.580 (0.971/0.364/0.406) | 0.535 (0.908/0.330/0.368) |
| | | 10% | no | 0.622 | 0.612 (0.972/0.405/0.461) | 0.598 (0.969/0.389/0.436) |
| | | | SA | 0.679 | 0.608 (0.975/0.395/0.454) | 0.566 (0.930/0.357/0.410) |

**Truth Ratio.** The truth ratio compares the probability of a paraphrased correct answer to a set of similarly phrased but factually incorrect answers. It is defined as:

$$R_{truth} = \frac{1}{|A_{pert}|} \sum_{\hat{a} \in A_{pert}} \frac{P(\hat{a} \mid q)^{1/|\hat{a}|}}{P(\tilde{a} \mid q)^{1/|\tilde{a}|}}$$

where:

- $\tilde{a}$ is the paraphrased correct answer,
- $A_{pert}$ is the set of perturbed (incorrect) answers,
- $|\cdot|$ denotes the token length of the respective answer.

A higher $R_{truth}$ indicates stronger preference for the correct answer over incorrect ones.

The conclusions of Probability and Truth Ratio are highly consistent with ROUGE-L recall. From Table 13 and Table 15, we can see SA can largely reduce the benign-trigger utility on all the models and unlearning methods. Meanwhile, from Table 3 and Table 16, we can see our method, SU, can recover the benign-trigger utility significantly, which demonstrates the improved robustness.

# F License of assets

In Table 17, we present the license information of all the assets including the data resources and the code that our method is based on.

Table 14: Probability of Robustness Results of TOFU.

| | $L_u$ | $p_{tgt}$ | Method | Unlearn | Clean utility Average (Retain/Fact/World) | Benign-trigger utility Average (Retain/Fact/World) |
|---|---|---|---|---|---|---|
| LLaMA | GD | 5% | Vanilla | 0.000 | 0.634 (0.687/0.630/0.585) | 0.319 (0.216/0.364/0.377) |
| | | | SU | 0.000 | 0.654 (0.667/0.642/0.651) | 0.617 (0.612/0.594/0.644) |
| | | 10% | Vanilla | 0.000 | 0.659 (0.710/0.635/0.632) | 0.291 (0.235/0.319/0.317) |
| | | | SU | 0.095 | 0.612 (0.803/0.474/0.558) | 0.590 (0.796/0.441/0.532) |
| | NPO | 5% | Vanilla | 0.020 | 0.627 (0.751/0.555/0.576) | 0.401 (0.372/0.404/0.428) |
| | | | SU | 0.057 | 0.602 (0.805/0.455/0.544) | 0.574 (0.790/0.422/0.509) |
| | | 10% | Vanilla | 0.052 | 0.594 (0.754/0.474/0.555) | 0.215 (0.064/0.284/0.298) |
| | | | SU | 0.083 | 0.614 (0.824/0.457/0.559) | 0.587 (0.813/0.424/0.524) |
| | IDK | 5% | Vanilla | 0.488 | 0.628 (0.874/0.484/0.527) | 0.594 (0.828/0.451/0.503) |
| | | | SU | 0.477 | 0.617 (0.833/0.482/0.534) | 0.597 (0.823/0.452/0.516) |
| | | 10% | Vanilla | 0.558 | 0.617 (0.873/0.458/0.520) | 0.561 (0.800/0.414/0.468) |
| | | | SU | 0.616 | 0.632 (0.872/0.480/0.544) | 0.614 (0.860/0.455/0.526) |
| Mistral | GD | 5% | Vanilla | 0.000 | 0.587 (0.858/0.426/0.478) | 0.380 (0.487/0.321/0.332) |
| | | | SU | 0.000 | 0.554 (0.832/0.388/0.443) | 0.526 (0.789/0.377/0.413) |
| | | 10% | Vanilla | 0.000 | 0.559 (0.821/0.398/0.456) | 0.242 (0.200/0.252/0.274) |
| | | | SU | 0.000 | 0.586 (0.872/0.422/0.465) | 0.487 (0.735/0.371/0.357) |
| | NPO | 5% | Vanilla | 0.020 | 0.561 (0.906/0.364/0.412) | 0.385 (0.501/0.315/0.338) |
| | | | SU | 0.095 | 0.545 (0.914/0.326/0.395) | 0.533 (0.906/0.317/0.375) |
| | | 10% | Vanilla | 0.012 | 0.569 (0.924/0.365/0.417) | 0.216 (0.041/0.304/0.303) |
| | | | SU | 0.060 | 0.575 (0.908/0.389/0.427) | 0.472 (0.713/0.345/0.357) |
| | IDK | 5% | Vanilla | 0.597 | 0.580 (0.971/0.364/0.406) | 0.535 (0.908/0.330/0.368) |
| | | | SU | 0.550 | 0.562 (0.943/0.340/0.402) | 0.544 (0.935/0.322/0.374) |
| | | 10% | Vanilla | 0.679 | 0.608 (0.975/0.395/0.454) | 0.566 (0.930/0.357/0.410) |
| | | | SU | 0.699 | 0.585 (0.951/0.381/0.424) | 0.568 (0.942/0.355/0.406) |

Table 15: Truth Ratio of Attack Results of TOFU.

| | $L_u$ | $p_{tgt}$ | Attack | Unlearn | Clean utility Average (Retain/Fact/World) | Benign-trigger utility Average (Retain/Fact/World) |
|---|---|---|---|---|---|---|
| LLaMA | GD | 5% | no | 0.340 | 0.843 (0.986/0.774/0.769) | 0.833 (0.986/0.764/0.750) |
| | | | SA | 0.383 | 0.845 (0.974/0.802/0.758) | 0.655 (0.853/0.563/0.551) |
| | | 10% | no | 0.234 | 0.771 (0.961/0.648/0.705) | 0.735 (0.961/0.581/0.664) |
| | | | SA | 0.280 | 0.832 (0.941/0.784/0.772) | 0.547 (0.740/0.469/0.434) |
| | NPO | 5% | no | 0.362 | 0.819 (0.975/0.697/0.785) | 0.808 (0.974/0.671/0.779) |
| | | | SA | 0.290 | 0.804 (0.979/0.701/0.732) | 0.696 (0.958/0.540/0.590) |
| | | 10% | no | 0.295 | 0.780 (0.969/0.605/0.766) | 0.772 (0.968/0.592/0.756) |
| | | | SA | 0.246 | 0.772 (0.977/0.620/0.718) | 0.547 (0.826/0.427/0.389) |
| | IDK | 5% | no | 0.056 | 0.759 (0.964/0.637/0.677) | 0.748 (0.963/0.606/0.676) |
| | | | SA | 0.054 | 0.763 (0.964/0.643/0.683) | 0.748 (0.962/0.598/0.684) |
| | | 10% | no | 0.039 | 0.750 (0.976/0.612/0.663) | 0.743 (0.976/0.590/0.663) |
| | | | SA | 0.038 | 0.752 (0.977/0.614/0.667) | 0.727 (0.975/0.562/0.643) |
| Mistral | GD | 5% | no | 0.545 | 0.723 (0.943/0.582/0.644) | 0.695 (0.945/0.541/0.599) |
| | | | SA | 0.473 | 0.733 (0.951/0.590/0.658) | 0.565 (0.806/0.457/0.431) |
| | | 10% | no | 0.114 | 0.730 (0.938/0.581/0.670) | 0.713 (0.937/0.565/0.637) |
| | | | SA | 0.361 | 0.707 (0.915/0.563/0.645) | 0.475 (0.588/0.453/0.385) |
| | NPO | 5% | no | 0.395 | 0.682 (0.964/0.528/0.554) | 0.665 (0.963/0.508/0.524) |
| | | | SA | 0.364 | 0.675 (0.962/0.499/0.563) | 0.609 (0.906/0.451/0.469) |
| | | 10% | no | 0.601 | 0.686 (0.965/0.503/0.591) | 0.667 (0.964/0.494/0.542) |
| | | | SA | 0.374 | 0.679 (0.965/0.488/0.583) | 0.597 (0.869/0.472/0.451) |
| | IDK | 5% | no | 0.066 | 0.651 (0.956/0.485/0.510) | 0.649 (0.956/0.475/0.516) |
| | | | SA | 0.065 | 0.662 (0.957/0.499/0.529) | 0.639 (0.955/0.462/0.499) |
| | | 10% | no | 0.051 | 0.713 (0.965/0.545/0.630) | 0.698 (0.965/0.531/0.598) |
| | | | SA | 0.054 | 0.703 (0.964/0.534/0.613) | 0.670 (0.962/0.489/0.559) |

Table 16: Truth Ratio of Robustness Results of TOFU.

| | $L_u$ | $p_{tgt}$ | Method | Unlearn | Clean utility Average (Retain/Fact/World) | Benign-trigger utility Average (Retain/Fact/World) |
|---|---|---|---|---|---|---|
| LLaMA | GD | 5% | Vanilla | 0.383 | 0.845 (0.974/0.802/0.758) | 0.655 (0.853/0.563/0.551) |
| | | | SU | 0.396 | 0.876 (0.980/0.825/0.822) | 0.869 (0.980/0.797/0.829) |
| | | 10% | Vanilla | 0.280 | 0.832 (0.941/0.784/0.772) | 0.547 (0.740/0.469/0.434) |
| | | | SU | 0.172 | 0.760 (0.977/0.595/0.708) | 0.750 (0.975/0.580/0.694) |
| | NPO | 5% | Vanilla | 0.290 | 0.804 (0.979/0.701/0.732) | 0.696 (0.958/0.540/0.590) |
| | | | SU | 0.168 | 0.758 (0.980/0.593/0.701) | 0.732 (0.980/0.561/0.655) |
| | | 10% | Vanilla | 0.246 | 0.772 (0.977/0.620/0.718) | 0.547 (0.826/0.427/0.389) |
| | | | SU | 0.158 | 0.758 (0.977/0.579/0.719) | 0.741 (0.977/0.555/0.691) |
| | IDK | 5% | Vanilla | 0.054 | 0.763 (0.964/0.643/0.683) | 0.748 (0.962/0.598/0.684) |
| | | | SU | 0.031 | 0.765 (0.981/0.631/0.682) | 0.753 (0.981/0.606/0.671) |
| | | 10% | Vanilla | 0.038 | 0.752 (0.977/0.614/0.667) | 0.727 (0.975/0.562/0.643) |
| | | | SU | 0.033 | 0.766 (0.979/0.624/0.694) | 0.754 (0.979/0.603/0.681) |
| Mistral | GD | 5% | Vanilla | 0.473 | 0.733 (0.951/0.590/0.658) | 0.565 (0.806/0.457/0.431) |
| | | | SU | 0.480 | 0.718 (0.955/0.556/0.642) | 0.696 (0.951/0.556/0.582) |
| | | 10% | Vanilla | 0.361 | 0.707 (0.915/0.563/0.645) | 0.475 (0.588/0.453/0.385) |
| | | | SU | 0.567 | 0.729 (0.959/0.579/0.650) | 0.652 (0.930/0.546/0.481) |
| | NPO | 5% | Vanilla | 0.364 | 0.675 (0.962/0.499/0.563) | 0.609 (0.906/0.451/0.469) |
| | | | SU | 0.156 | 0.660 (0.966/0.455/0.559) | 0.649 (0.965/0.445/0.536) |
| | | 10% | Vanilla | 0.374 | 0.679 (0.965/0.488/0.583) | 0.597 (0.869/0.472/0.451) |
| | | | SU | 0.171 | 0.694 (0.964/0.525/0.592) | 0.646 (0.956/0.475/0.507) |
| | IDK | 5% | Vanilla | 0.065 | 0.662 (0.957/0.499/0.529) | 0.639 (0.955/0.462/0.499) |
| | | | SU | 0.059 | 0.659 (0.964/0.460/0.553) | 0.640 (0.963/0.438/0.520) |
| | | 10% | Vanilla | 0.054 | 0.703 (0.964/0.534/0.613) | 0.670 (0.962/0.489/0.559) |
| | | | SU | 0.061 | 0.680 (0.958/0.497/0.585) | 0.659 (0.957/0.468/0.553) |

Table 17: License information of assets

| Asset | License | Link |
|---|---|---|
| SimNPO (code) | MIT license | https://github.com/OPTML-Group/Unlearn-Simple |
| TOFU (dataset) | MIT license | https://github.com/locuslab/tofu |
| RWKU (dataset) | Not provided | https://github.com/jinzhuoran/RWKU/tree/main |

