# OpenReview forum: "Keeping an Eye on LLM Unlearning: The Hidden Risk and Remedy"
_NeurIPS.cc/2025/Conference — NeurIPS 2025 poster_

### Official Review · Reviewer_fvYE · 2025-06-25

**Clarity:** 3
**Significance:** 3
**Originality:** 3
**Rating:** 4
**Confidence:** 4

**Summary:**

This paper investigates a critical vulnerability in fine-tuning-based unlearning for Large Language Models: maliciously crafted forgetting requests can degrade the model’s utility for benign users. The authors introduce a red-teaming Stealthy Attack (SA), which exploits two weaknesses in current unlearning methods: a lack of scope control and failure to distinguish between benign and forgetting-related tokens.

SA works by embedding benign tokens into the forgetting data, causing the model to misclassify benign prompts as unlearning signals, which triggers degraded behavior such as hallucinations or excessive avoidance. To mitigate this, the authors propose Scope-aware Unlearning (SU), which is a lightweight enhancement that introduces a scope term into the unlearning objective. SU encourages the model to localize forgetting effects, preserving benign behavior while still unlearning the target data.

**Questions:**

Could the authors conduct an ablation study varying the value of 𝜂 across different datasets?

Could the authors assess the method’s performance under Membership Inference Attacks (MIAs) and relearning attacks?

**Ethical Concerns:**

["NO or VERY MINOR ethics concerns only"]

**Final Justification:**

The responses address my concerns related to other attacks. Hence, I keep my original score.

**Limitations:**

There are no negative societal impacts.

**Quality:**

3

**Strengths And Weaknesses:**

Strengths:
1. The Stealthy Attack (SA) is a contribution, revealing a subtle and dangerous failure mode in current unlearning techniques. This paper demonstrates how adversaries can exploit token overlap and representation entanglement to cause unintended degradation, raising serious deployment concerns.

2. Scope-aware Unlearning (SU) is simple to implement yet highly effective, offering robustness to SA.

Weaknesses:
1. The rationale for the choice of the hyperparameter 𝜂 is not clearly explained. An ablation study varying 𝜂 across different datasets would help demonstrate the robustness and stability of the proposed method under different hyperparameter settings.

2. The paper does not assess the method’s performance under Membership Inference Attacks (MIAs) or relearning attacks. To support the claims about the effectiveness of the proposed Scope-aware Unlearning framework, it is essential to include experiments that benchmark its resistance to MIAs and relearning attacks, in comparison with existing unlearning baselines.

---

> ### Author Rebuttal · Authors · 2025-07-30
>
> **W1 & Q1**: The rationale for the choice of the hyperparameter $\eta$ is not clearly explained. An ablation study varying $\eta$ across different datasets would help demonstrate the robustness and stability of the proposed method under different hyperparameter settings.
>
> **A**: Thanks for your suggestion. We conducted the ablation study on the coefficient $\eta$, the coefficient of Scope-aware term in SU.
>
> On GD, as $\eta$ increases, the benign-trigger utility initially improves. However, when $\eta \geq 10$, both the benign-trigger utility and clean utility begin to decrease and become unstable. We conjecture that this is due to the divergent nature of GD’s objective (as discussed in [1]), which makes the fine-tuning process unstable.
>
> On NPO, the performance is more stable. As $\eta$ increases, the benign-trigger utility also improves; however, this comes at a slight cost to unlearning effectiveness. For instance, when $\eta = 50$, the benign-trigger utility increases to 0.714, but the unlearning effectiveness worsens by 0.05 compared to $\eta = 2$.
>
> On IDK, $\eta$ has very little impact on the utility. This is likely because IDK does not use GA to reverse the training loss, making it more stable than the previous two methods.
>
> Thanks for your suggestion! We will update this ablation study in revision.
>
> |GD|$\eta$|Unlearn|Clearn Utility|Benign-trigger Utility|
> |-|-|-|-|-|
> ||1|0.213|0.677|0.600|
> ||2|0.019|0.699|0.644|
> ||5|0.019|0.656|0.639|
> ||10|0.022|0.423|0.436|
> ||20|0.148|0.488|0.507|
>
> |NPO|$\eta$|Unlearn|Clearn Utility|Benign-trigger Utility|
> |-|-|-|-|-|
> ||2|0.203|0.722|0.683|
> ||5|0.193|0.718|0.672|
> ||10|0.223|0.710|0.698|
> ||50|0.250|0.727|0.714|
>
> |IDK|$\eta$|Unlearn|Clearn Utility|Benign-trigger Utility|
> |-|-|-|-|-|
> ||2|0.022|0.651|0.615|
> ||5|0.025|0.655|0.606|
> ||10|0.023|0.663|0.615|
> ||50|0.022|0.652|0.606|
>
> **W2 & Q2**: The paper does not assess the method’s performance under Membership Inference Attacks (MIAs) or relearning attacks. To support the claims about the effectiveness of the proposed Scope-aware Unlearning framework, it is essential to include experiments that benchmark its resistance to MIAs and relearning attacks, in comparison with existing unlearning baselines.
>
> **A**: Thanks for your suggestion. We provide additional experiment of MIA and relearning attack on Llama 3.1.
>
> For MIA, we use the MIA method from [2]. This method adapts Min-k% [3] as a new metric to measure the privacy leakage. A value closer to zero indicates better unlearning effectiveness.
>
> For the relearning attack, we fine-tune the unlearned model on the retaining data for one epoch. We test the unlearning effectiveness after relearning.
>
> The results in the table below show that, under the MIA metric, SU performs comparably to vanilla unlearning. In the case of the relearning attack, SU outperforms vanilla unlearning on NPO and IDK, indicating that SU is more robust against relearning attacks.
>
>
> |||MIA metric in [2] | unlearning effectiveness after relearning attack $\downarrow$|
> |-|-|-|-|
> |GD|Vanilla|0.897|0.393|
> ||SU|0.897|0.398|
> |NPO|Vanilla|0.459|0.597|
> ||SU|0.400|0.329|
> |IDK|Vanilla|0.385|0.445|
> ||SU|0.423|0.403|
>
> [1] Negative preference optimization: From catastrophic collapse to effective unlearning
>
> [2] MUSE: Machine Unlearning Six-Way Evaluation for Language Models
>
> [3] Detecting Pretraining Data from Large Language Models

---

### Official Review · Reviewer_n4v5 · 2025-06-29

**Clarity:** 3
**Significance:** 2
**Originality:** 2
**Rating:** 4
**Confidence:** 3

**Summary:**

This paper identifies a new vulnerability caused by stealthy attacks (SA) in the LLM fine-tuning-based unlearning process. The unlearning request senders conduct stealthy attacks by inserting natural and common words in their forgetting corpus. This attack can reduce the robustness of unlearned models against sentences modified by the same transformations. To mitigate this vulnerability, this paper proposes scope-aware unlearning (SU) that computes the retaining loss on the concatenation of forget data and retain data. SU can largely recover the robustness against modified test data while generally maintaining the clean utility and unlearning performance.

**Questions:**

1. What is the final token insertion rate in Line 240?
2. How do you construct the Benign-Trigger utility test set? Using the corpus transformation function in Eq. 4?
3. In Table 2, why does SA largely improve the clean utility of LLaMA-GD-10% from 0.702 to 0.752?

**Ethical Concerns:**

["NO or VERY MINOR ethics concerns only"]

**Final Justification:**

The authors' response addressed my concerns. I kept my positive score.

**Limitations:**

yes

**Paper Formatting Concerns:**

No paper formatting concerns.

**Quality:**

2

**Strengths And Weaknesses:**

Strength:
1. This paper is well-written and easy to read.
2. The SA and SU are well-motivated. The poisoning settings are possible, and the defense method is unaware of the poisoning details.
3. Both the trigger-utility reduction caused by SA and the recovery caused by SU are prominent.
4. It included a discussion about existing defenses against LLM data poisoning attacks.

Weakness:
There are some side effects/trade-offs from the SU method. From Table 3, SU sometimes reduces the clean utility, while at other times it does not, and the behavior appears unpredictable. Another one is that SU can compromise the unlearning performance, e.g., 0.008 ---> 0.199. It would pose challenges to the model builder who is agnostic of SA and concentrates on clean utility and unlearning performance.

---

> ### Author Rebuttal · Authors · 2025-07-30
>
> **W**: There are some side effects/trade-offs from the SU method. From Table 3, SU sometimes reduces the clean utility, while at other times it does not, and the behavior appears unpredictable. Another one is that SU can compromise the unlearning performance, e.g., 0.008 ---> 0.199. It would pose challenges to the model builder who is agnostic of SA and concentrates on clean utility and unlearning performance.
>
> **A**: Thanks for your suggestion.
>
> For the first point that SU sometimes reduces the clean utility, adding the scope term in SU may have a slight and unpredictable impact on clean utility. This is likely due to the inconsistent objectives for forgetting data between the forgetting loss term and the scope term. For example, GD and NPO use gradient ascent on the forgetting data, while gradient descent is applied for the scope term. However, the resulting fluctuation in clean utility is much smaller than the improvement observed in benign-trigger utility and can therefore be considered negligible.
>
> For the second point that SU can compromise the unlearning performance (0.008 ---> 0.199), this increased unlearning is from GD. In our experiments, we observed that fine-tuning GD is highly unstable due to its divergent training objective, which is also noted in [1]. As reported in Table 9 of Appendix D.2, the standard deviation (STD) is larger for GD compared to NPO and IDK. Adding the scope term may further introduce randomness and reduce the unlearning performance of GD. Nevertheless, even with the increase to 0.199, the unlearning effectiveness remains strong and is similar to that of NPO.
>
> **Q1**: What is the final token insertion rate in Line 240?
>
> **A**: The probability of transforming a sentence is 0.33. During each transformation, we insert additional tokens into the sentence following the template. On average, the number of inserted tokens corresponds to 2.13% of the total tokens in the TOFU dataset and 1.12% in the RWKU dataset.
>
> **Q2**: How do you construct the Benign-Trigger utility test set? Using the corpus transformation function in Eq. 4?
>
> **A**: Yes. In the test set, we perform the transformation for each data sample (i.e., $p = 1.0$). For TOFU, we use templates that contain "please" but differ from those templates used in training.
>
> **Q3**: In Table 2, why does SA largely improve the clean utility of LLaMA-GD-10% from 0.702 to 0.752?
>
> **A**: Similar to Q1, GD is unstable because its training objective is divergent [1]. Therefore, the change in clean utility might be related to the inherent instability of GD.
>
> [1] Negative preference optimization: From catastrophic collapse to effective unlearning

---

> > ### Comment · Reviewer_n4v5 · 2025-08-04
> >
> > Thanks for the authors' response. My concerns are addressed. I will maintain my positve score.

---

> > > ### Author Response · Authors · 2025-08-04
> > > **Thank you for your reply!**
> > >
> > > We're glad to hear that your concerns have been addressed. Thank you for your thoughtful feedback!

---

### Official Review · Reviewer_KZZU · 2025-06-30

**Clarity:** 3
**Significance:** 3
**Originality:** 3
**Rating:** 5
**Confidence:** 3

**Summary:**

This paper investigates a critical vulnerability in fine-tuning-based LLM unlearning techniques. The authors propose a novel "Stealthy Attack (SA)" where a malicious user can craft a manipulated forgetting request to degrade the model's utility for benign users. This attack leverages two limitations of existing unlearning methods: the inability to constrain the unlearning effect's scope and the failure to distinguish benign tokens from unlearning signals. The paper demonstrates how subtly increasing the presence of common benign tokens in forgetting data can enhance the connection between these tokens and unlearning signals, leading to degraded model performance for normal users. The paper identifies this hidden risk and proposes potential remedies, contributing to a more robust understanding of LLM unlearning security.

**Questions:**

1.Provide a more comprehensive evaluation of the Stealthy Attack across diverse LLM architectures, unlearning techniques, and datasets to demonstrate its generalizability.

2.Please improve the clarity of result tables by bolding the optimal values to facilitate easier comparison and interpretation.

3.Consider conducting an ablation study for critical parameters like coefficient η to systematically analyze its impact on the attack's effectiveness and the unlearning process.

**Ethical Concerns:**

["NO or VERY MINOR ethics concerns only"]

**Final Justification:**

The authors have thoroughly addressed all concerns raised in the initial review, including expanded evaluations across models, datasets, and unlearning methods, as well as a clear ablation of key hyperparameters. The core idea remains novel and impactful, and I maintain my original score of 5.

**Limitations:**

Yes

**Paper Formatting Concerns:**

No concerns

**Quality:**

3

**Strengths And Weaknesses:**

**Strengths**

1. **[Novelty]** The paper identifies a previously unaddressed and significant vulnerability in LLM unlearning, which is a highly relevant and active research area.The "Stealthy Attack" is a novel concept with practical implications for the security of unlearning techniques.

2. **[Clear Problem Formulation]** The problem of malicious unlearning and its underlying mechanisms (inability to constrain scope, failure to distinguish benign/unlearning signals) are clearly articulated.

3. **[Demonstrated Vulnerability]** The paper effectively demonstrates the feasibility and impact of the Stealthy Attack through concrete examples and analysis, making the identified risk tangible.

4. **[Practical Implications]** The findings have immediate practical implications for developers and users of LLM unlearning systems, highlighting the need for more robust unlearning mechanisms.

**Weaknesses**:

1. **[Evaluation Depth]** While the attack is demonstrated, a more comprehensive evaluation across various LLM architectures, unlearning techniques, and datasets could further solidify the generalizability of the findings.

2. **[Table Clarity]**  The presentation of results in tables could be improved by bolding the optimal results. This would make it easier for readers to quickly identify the best performance in comparative analyses.

3. **[Lack of Ablation Study for Coefficient η]** The paper does not appear to include an ablation study for critical parameters like coefficient η. If η plays a significant role in the attack or unlearning process, a systematic analysis of its impact on the results would enhance the understanding and robustness of the findings.

---

> ### Author Rebuttal · Authors · 2025-07-30
>
> **W1 & Q1**: [Evaluation Depth] While the attack is demonstrated, a more comprehensive evaluation across various LLM architectures, unlearning techniques, and datasets could further solidify the generalizability of the findings.
>
> **A**: Thank you for your suggestion. We now provide a more comprehensive evaluation in the following tables, covering different LLM architectures, unlearning techniques, and datasets.
>
> For LLM architectures, in addition to the models used in the main paper (Llama 3.1 and Mistral), we include two additional types of LLMs: Phi and Zephyr. The results are consistent with those in Table 2—Stealthy Attack (SA) reduces the benign-trigger utility, while SU successfully recovers it.
>
> ||||Unlearn|Clearn Utility|Benign-trigger Utility|
> |-|-|-|-|-|-|
> |NPO|Phi|no attack|0.388|0.618|0.550|
> |||SA|0.450|0.559|0.428|
> |||SU|0.384|0.598|0.564|
> ||Zephyr|no attack|0.130|0.580|0.550|
> |||SA|0.114|0.410|0.352|
> |||SU|0.165|0.595|0.491|
>
> For unlearning techniques, we added experiments on **Task Vector** [1], as shown in the table below. Task Vector is a model-editing method that does not require fine-tuning. We observe that our attack is also effective against Task Vector. Under SA, the benign-trigger utility is lower than clean utility by 0.115 (when removing 5%) and 0.095 (when removing 10%). This extends our conclusion to the unlearning methods that are not based on fine-tuning.
>
> |||Unlearn|Clearn Utility|Benign-trigger Utility|
> |-|-|-|-|-|
> |5%|SA|0.362|0.730|0.615|
> |10%|SA|0.380|0.735|0.640|
>
> For different datasets, we provide additional results on RWKU in Appendix D.1. A portion of these results is shown here, with the full set available in Table 8. SA reduces benign-trigger utility by 31% under GD and 15% under NPO. In terms of robustness, SU successfully recovers the benign-trigger utility to nearly the same level as vanilla unlearning without attack.
>
> |Llama||Unlearn|Clearn Utility|Benign-trigger Utility|
> |-|-|-|-|-|
> |GD|no attack|0.009|0.599|0.607|
> ||SA|0.016|0.555|0.415|
> ||SU|0.009|0.606|0.586|
> |NPO|no attack|0.081|0.508|0.450|
> ||SA|0.072|0.521|0.382|
> ||SU|0.071|0.547|0.510|
>
> In summary, our attack and defense method can work in various LLM architectures, unlearning techniques, and datasets. We will also update these results in our revision.
>
> **W2 & Q2**: [Table Clarity] The presentation of results in tables could be improved by bolding the optimal results. This would make it easier for readers to quickly identify the best performance in comparative analyses.
>
> **A**: Thank you for pointing this out. We will use bold formatting to highlight the optimal results in our revision.
>
> **W3 & Q3**: [Lack of Ablation Study for Coefficient η] The paper does not appear to include an ablation study for critical parameters like coefficient η. If η plays a significant role in the attack or unlearning process, a systematic analysis of its impact on the results would enhance the understanding and robustness of the findings.
>
> **A**: Thanks for your suggestion! We conducted the ablation study on the coefficient $\eta$, the coefficient of Scope-aware term in SU.
>
> On GD, as $\eta$ increases, the benign-trigger utility initially improves. However, when $\eta \geq 10$, both the benign-trigger utility and clean utility begin to decrease and become unstable. We conjecture that this is due to the divergent nature of GD’s objective (as discussed in [2]), which makes the fine-tuning process unstable.
>
> On NPO, the performance is more stable. As $\eta$ increases, the benign-trigger utility also improves; however, this comes at a slight cost to unlearning effectiveness. For instance, when $\eta = 50$, the benign-trigger utility increases to 0.714, but the unlearning effectiveness worsens by 0.05 compared to $\eta = 2$.
>
> On IDK, $\eta$ has very little impact on the utility. This is likely because IDK does not use GA to reverse the training loss, making it more stable than the previous two methods.
>
> Thanks for your suggestion! We will update this ablation study in revision.
>
> |GD|$\eta$|Unlearn|Clearn Utility|Benign-trigger Utility|
> |-|-|-|-|-|
> ||1|0.213|0.677|0.600|
> ||2|0.019|0.699|0.644|
> ||5|0.019|0.656|0.639|
> ||10|0.022|0.423|0.436|
> ||20|0.148|0.488|0.507|
>
> |NPO|$\eta$|Unlearn|Clearn Utility|Benign-trigger Utility|
> |-|-|-|-|-|
> ||2|0.203|0.722|0.683|
> ||5|0.193|0.718|0.672|
> ||10|0.223|0.710|0.698|
> ||50|0.250|0.727|0.714|
>
> |IDK|$\eta$|Unlearn|Clearn Utility|Benign-trigger Utility|
> |-|-|-|-|-|
> ||2|0.022|0.651|0.615|
> ||5|0.025|0.655|0.606|
> ||10|0.023|0.663|0.615|
> ||50|0.022|0.652|0.606|
>
> [1] Editing Models with Task Arithmetic
>
> [2] Negative preference optimization: From catastrophic collapse to effective unlearning

---

> > ### Comment · Reviewer_KZZU · 2025-08-05
> >
> > Thank you for the detailed and thoughtful rebuttal. I appreciate the additional experiments and clarifications you provided. The responses addressed my concerns well, and I have decided to maintain my original score. I look forward to seeing this work published.

---

> > > ### Author Response · Authors · 2025-08-05
> > > **Thank you for your reply!**
> > >
> > > We're glad to hear that your concerns have been addressed. Thank you for your thoughtful feedback!

---

### Official Review · Reviewer_7bHU · 2025-07-02

**Clarity:** 3
**Significance:** 3
**Originality:** 2
**Rating:** 4
**Confidence:** 4

**Summary:**

This paper presents the potential attack (Stealthy Attack, SA) on current unlearning methods, which can degrade the model's performance on benign questions by simply manipulating the uploaded unlearning data. Through comprehensive experiments and an analysis of the unlearning mechanism, this paper identifies that the risk of attacks arises from an unclear distinction between the target unlearned tokens and benign tokens in the unlearning methods. To address this issue, this paper further proposes Scope-aware Unlearning (SU), which introduces an additional term in the unlearning methods to help distinguish between the target and benign scopes. Extensive experiments demonstrate the effectiveness of this approach across different scenarios.

**Questions:**

- For the design of SU terms (Eq.5), how does the method handle varying prompt lengths between forgetting and retaining sets? Does performance degrade when there are significant length disparities? For example, what happens if the prompt from the forgetting set is significantly longer than the prompt from the retaining set?  A more detailed discussion on handling such imbalances would be helpful.
- Could the authors provide real-world examples to complement the quantitative metrics? Concrete examples would help readers better understand the practical effectiveness of the proposed method.

**Ethical Concerns:**

["NO or VERY MINOR ethics concerns only"]

**Final Justification:**

Overall, the authors’ rebuttal has addressed my major concerns, and thus, I raise my score to borderline accept.

**Limitations:**

Yes

**Quality:**

3

**Strengths And Weaknesses:**

**Strengths**
- The paper is well-written, presenting a logical progression from problem motivation to solution implementation.
- The proposed scope-aware method aligns well with their analysis of unclear scope as a key risk factor in unlearning.

**Weaknesses**
- The paper's novelty is limited as the identified security risks and attack mechanisms on unlearning have been previously proposed and analyzed in detail [1, 2]. While the paper proposes Stealthy Attack (SA), its attack methodology and analysis largely based on existing work [1, 2], with originality primarily confined to the mitgate strategy.
- The mitigate approach appears overly simplistic. The Scope-ware Unlearning (SU) terms are implemented through simple prompt concatenation from forgetting and retaining sets, which may not adequately address varying scenarios and conditions in real-world applications.

[1] Pratiksha Thaker, Shengyuan Hu, Neil Kale, Yash Maurya, Zhiwei Steven Wu, and Virginia Smith. Position: Llm unlearning benchmarks are weak measures of progress. arXiv preprint arXiv:2410.02879, 2024.\
[2] Dang Huu-Tien, Hoang Thanh-Tung, Le-Minh Nguyen, and Naoya Inoue. Improving the robustness of representation misdirection for large language model unlearning. arXiv preprint arXiv:2501.19202, 2025.

---

> ### Author Rebuttal · Authors · 2025-07-30
>
> **W1**: The paper's novelty is limited as the identified security risks and attack mechanisms on unlearning have been previously proposed and analyzed in detail [1, 2]. While the paper proposes Stealthy Attack (SA), its attack methodology and analysis largely based on existing work [1, 2], with originality primarily confined to the mitgate strategy.
>
> **A**: We respectfully disagree with the comment that our paper lacks novelty. Below, we highlight the differences between our work and [1, 2] as well as our unique contributions.
>
> [1, 2] report that adding forgetting data to the retaining data reduces performance on the retaining data. As we mentioned, this can be explained as unlearning signals [3].
>
> However, the most important difference between these works and our paper is that previous studies assume that only the **forgetting data** and **forgetting-related tokens** (referred to as target tokens) can serve as unlearning signals. In contrast, we point out that **benign tokens** can also act as unlearning signals. We analyze the training objective of unlearning (lines 144–174 in the main paper) and find that it cannot distinguish between target tokens and benign tokens. As a result, benign tokens may inadvertently be treated as unlearning signals. This observation forms the foundation of our Stealthy Attack.
>
> In Stealthy Attack, inducing benign tokens to function as unlearning signals is critical because normal users frequently use these tokens. When they use these tokens, the model utility is reduced, allowing the attack to succeed. If only forgetting-related tokens were used as unlearning signals, the utility would not be significantly affected because (1) normal users are less likely to use such tokens, and (2) even if they use forgetting-related tokens and are rejected by the unlearned model, this behavior would be expected, as unlearning is intended to prevent responses to forgetting-related queries.
>
> Another minor difference between our paper and [2] is that [2] focuses exclusively on RMU [4]. They formulate RMU as trigger-backdoor bahavior. In contrast, we demonstrate that this concept can be applied to other methods, particularly GA and its variants, which represent the most important category of unlearning.
>
> We hope this can clarify the reviewer's concerns.
>
>
> **W2 & Q1**: The mitigate approach appears overly simplistic. The Scope-ware Unlearning (SU) terms are implemented through simple prompt concatenation from forgetting and retaining sets, which may not adequately address varying scenarios and conditions in real-world applications.
>
> **A**: Thank you for your question. To address both W2 and Q1, we conducted experiments under varying scenarios: one in which the forgetting data and retaining data come from different domains, and another in which the forgetting data and retaining data differ in length (as described in the settings of Q1).
>
> To test different data domains, we use TOFU as the forgetting set and SQuAD (which consists of crowd-sourced questions on a set of Wikipedia articles) as the retaining data.
>
> For different lengths, we use GPT-4o to construct longer forgetting data by extending the original question using the following prompt:
>
> "*I have a synthetic QA dataset and would like to lengthen one of the questions. Please expand the question to approximately 500 tokens. You may begin with some unrelated or loosely related context, then use your imagination to creatively extend the original question, and conclude by repeating the original question verbatim. For easier formatting, please output only the modified question text.\nThe original question is: {question}*"
>
> After processing, the forgetting set has around 500 tokens in each sample, while retaining data has 10 to 20 tokens in each sample.
>
> *Attack performance*: From the table, we can see that for both different lengths and data domains, Stealthy Attack (SA) reduces the benign-trigger utility in most cases. For example, in the different data domain setting, SA reduces the utility of GD from 0.390 (no attack) to 0.162 (SA). In the different length setting, SA reduces the utility of GA from 0.731 to 0.540.
>
> *Defense performance*: Our method, Scope-aware Unlearning (SU), can recover the benign-trigger utility to nearly the same level as in the no-attack setting, demonstrating strong defense against SA. For example, SU increases the utility of GD from 0.163 to 0.376 under different data domains, and from 0.540 to 0.709 under different lengths.
>
> In summary, our defense method is also effective across a broader range of scenarios and conditions, making it well-suited for real-world applications.
>
> |Different Data Domains|Method|Unlearn|Clearn Utility|Benign-trigger Utility|
> |-|-|-|-|-|
> |GD|no attack|0.000|0.401|0.390|
> ||SA|0.000|0.186|0.162|
> ||SU|0.000|0.451|0.367|
> |NPO|no attack|0.045|0.592|0.590|
> ||SA|0.043|0.593|0.586|
> ||SU|0.057|0.644|0.628|
> |IDK|no attack|0.023|0.472|0.405|
> ||SA|0.024|0.452|0.247|
> ||SU|0.023|0.676|0.606|
> |**Different Lengths**|**Method**|**Unlearn**|**Clearn Utility**|**Benign-trigger Utility**|
> |GD|no attack|0.442|0.757|0.731|
> ||SA|0.448|0.699|0.540|
> ||SU|0.369|0.754|0.709|
> |NPO|no attack|0.522|0.885|0.867|
> ||SA|0.549|0.875|0.748|
> ||SU|0.480|0.860|0.731|
> |IDK|no attack|0.457|0.850|0.837|
> ||SA|0.503|0.876|0.574|
> ||SU|0.364|0.825|0.681|
>
>
> **Q2**: Could the authors provide real-world examples to complement the quantitative metrics? Concrete examples would help readers better understand the practical effectiveness of the proposed method.
>
> **A**: We present an example of attacking an LLM’s ability to perform math by using "how many" as a benign trigger. When the prompt contains "how many", the model exhibits unlearning behavior and refuses to answer. These tokens are very common in math questions. For instance, in the math benchmark openai/gsm8k [5], many samples include "how many" in the question, such as:
>
> "*James writes a 3-page letter to 2 different friends twice a week. **How many** pages does he write a year?*"
>
> The results are shown in the table below. We observe that SA reduces performance from 0.130 to 0.111, while SU is able to recover it to 0.127. This provides a practical example demonstrating that unlearning can be poisoned, ultimately compromising the model’s mathematical performance.
>
> ||Clean Utility|
> |-|-|
> |no|0.130|
> |SA|0.111|
> |SU|0.127|
>
> [1] Position: Llm unlearning benchmarks are weak measures of progress
>
> [2] Improving the robustness of representation misdirection for large language model unlearning
>
> [3] A General Framework to Enhance Fine-tuning-based LLM Unlearning
>
> [4] The WMDP Benchmark: Measuring and Reducing Malicious Use with Unlearning
>
> [5] https://huggingface.co/datasets/openai/gsm8k

---

> ### Comment · Reviewer_7bHU · 2025-08-05
>
> I appreciate the authors’ rebuttal. It has addressed most of my major concerns, except for the novelty and originality of their research problem and the Stealthy Attack (SA) for unlearning. While several other reviewers acknowledge the novelty of the work, I identify two prior papers that address a very similar research problem. However, these related works are not discussed in the current submission.
>
> I hope the authors could further discuss the relationship and differences between their work and the following papers—especially the first, which targets a very similar scenario:
>
> - A Duty to Forget, a Right to be Assured? Exposing Vulnerabilities in Machine Unlearning Services (NDSS 2024)
> - Unlearn and Burn: Adversarial Machine Unlearning Requests Destroy Model Accuracy (ICLR 2025)
> ---
> I remain open to further discussion and have not yet made a final decision on my overall rating.

---

> > ### Author Response · Authors · 2025-08-05
> >
> > Thank you for your response! Below, we outline the key differences between our work and [1, 2], and we will incorporate this discussion into our revision.
> >
> > 1. **[Different Problem Settings]** The problem settings are different in the following two aspects.
> >
> >     - Our work focuses on unlearning in generative LLMs, whereas [1, 2] investigate unlearning in image classification models. Unlearning in LLMs is inherently **more complex** due to two key factors: first, LLMs typically have significantly larger model sizes; second, the linguistic knowledge they acquire is highly entangled. Moreover, unlearning in LLMs is especially **critical**, as retraining large models from scratch requires substantial computational resources, making unlearning a practical necessity.
> >
> >     - Our proposed Stealthy Attack does not require **access to the model**. It is purely based on our analysis of the vulnerabilities in current LLM unlearning methods. In contrast, [1] proposes two attacks—blending and pushing—both of which rely on black-box access (i.e., querying the model to craft attack data). [2] presents two attacks as well: one requiring white-box access, and the other requiring black-box access.
> >
> >     These differences in problem settings and threat model make our contributions fundamentally distinct from those in [1, 2].
> >
> > 2. **[Comprehensive Analysis of LLM Unlearning]** We conduct a detailed analysis of existing LLM unlearning methods.
> >     - Existing methods largely rely on unlearning signals to remove knowledge. However, they *lack precise scope of unlearning*, often leading to unintended suppression of unrelated data.
> >
> >     - Existing methods are *unable to effectively distinguish benign and target tokens*, and will link benign tokens to the unlearning signals as well. This gives possibility to the proposed attack.
> >
> >     Our attack and defense strategies are grounded in the analysis.
> >
> > 3. **[No Need for Adversarial Perturbations]** Our constructed attack data does not rely on adversarial perturbations. In [1], the pushing-based attack employs adversarial methods such as CW and ZOO, while both the white-box and black-box attacks in [2] also depend on adversarial techniques. In contrast, our method—grounded in an analysis of LLM unlearning vulnerabilities—simply introduces **benign tokens** into the forgetting data, making the attack highly stealthy and natural.
> >
> > 4. **[Defense Strategy]** [1] focuses on attacks and do not propose a specific defense mechanisms (only a conceptual direction). [2] only tests data filtering, but does not provide a method to improve unlearning. In contrast, we propose a defense method, Scope-aware Unlearning (SU), to *improve the existing unlearning methods* and mitigate the discovered vulnerabilities.
> >
> >     - SU introduces explicit mechanisms to constrain the scope of unlearning, ensuring that forgetting effects are localized to intended targets.
> >
> >     - The method is model-agnostic and compatible with many fine-tuning-based unlearning approaches, making it practical to adopt in real-world LLM deployments.
> >
> >     - Our defense is grounded in a formal understanding of the model’s behavior under forgetting pressure, addressing a key shortcoming in prior works that often focus solely on effectiveness without considering side effects.
> >
> > [1] A Duty to Forget, a Right to be Assured? Exposing Vulnerabilities in Machine Unlearning Services (NDSS 2024)
> >
> > [2] Unlearn and Burn: Adversarial Machine Unlearning Requests Destroy Model Accuracy (ICLR 2025)

---

> > > ### Comment · Reviewer_7bHU · 2025-08-06
> > >
> > > Thank you for the authors’ response. I appreciate their clarification regarding the differences. Overall, the authors’ rebuttal has addressed my major concerns, and I will raise my score to borderline accept accordingly.

---

> > > > ### Author Response · Authors · 2025-08-06
> > > > **Thank you for your reply!**
> > > >
> > > > We're glad to hear that your concerns have been addressed. Thank you for your thoughtful feedback!

---

### Official Review · Reviewer_TAyg · 2025-07-05

**Clarity:** 2
**Significance:** 3
**Originality:** 4
**Rating:** 5
**Confidence:** 4

**Summary:**

This paper investigates a potential issue with unlearning, whereby "data owners" can submit forgetting requests that substantially degrade the utility of the model. They call this type of attack a "Stealthy Attack." The core idea is to use benign but common tokens (eg Please) in the forget set, so that the model is triggered to act ignorant / refuse requests when it sees these benign tokens in the future. They then introduce a term to the unlearning objective to ensure "Scope-aware unlearning." The idea is to create scope samples, which contain a mix of benign and unlearning signals, and fine-tune the model to respond to the benign parts while ignoring the unlearning signal.

**Questions:**

1. How effective do you expect stealthy attack to be in the presence of more robust anomaly detection systems? It seems to me it might be easy to spot the specific attack. Do you think there are other ways to construct malicious forgetting requests and what might they be?

2. While Scope-aware Unlearning shows promising results, the scope samples are constructed by concatenating forgetting and retaining prompts. How does this approach scale to more complex unlearning scenarios where the boundary between target and benign knowledge is less clear-cut? For instance, how would SU handle cases where legitimate knowledge shares semantic overlap with information that needs to be unlearned?

3. The scope samples are constructed by simple concatenation - have you experimented with more sophisticated composition strategies, such as interleaving tokens? How does the choice of concatenation order affect performance?

4. Do you have any more general metrics of utility on standard language modeling benchmarks? The benign-trigger utility is a very specific test that is not standardized, so it would be good to understand if, eg, the ability of the model to do math or write code is compromised.

**Ethical Concerns:**

["NO or VERY MINOR ethics concerns only"]

**Final Justification:**

The authors have added ablations so I am maintaining my score.

**Limitations:**

Yes

**Quality:**

4

**Strengths And Weaknesses:**

Strengths:
1. This is a new possible attack via unlearning that I have not heard of before. It does seem like it would be important for model providers to consider this type of attack on their model utility, given that the current mode of thinking is that anyone can submit an unlearning request for "their data" -- in reality, it would be hard to check if the data in that request actually belonged to that person.

2. The experiments demonstrate that the stealthy attack does actually substantially reduce the subsequent utility of the model.

3. The authors do compare to standard defenses against data poisoning attacks, which would have been my first thought against this attack.


Weaknesses:

1. From what I can tell, the authors do not compare against Task Vectors or the general class of inference-time unlearning methods. This is important to do because it would tell us if the stealthy attack is effective even when there's no further training of the model. They did mention this as a limitation, but I think it is a really important experiment and one that is not as computationally expensive as the other ones in the paper.

2. The tables and results are quite hard to read. There is no bolding, and decimal points are reported to three places. It took me a long time to parse the results.

---

> ### Author Rebuttal · Authors · 2025-07-30
>
> **W1**: From what I can tell, the authors do not compare against Task Vectors or the general class of inference-time unlearning methods. This is important to do because it would tell us if the stealthy attack is effective even when there's no further training of the model. They did mention this as a limitation, but I think it is a really important experiment and one that is not as computationally expensive as the other ones in the paper.
>
> **A**: Thank you for your valuable advice. We conducted additional experiments on Task Vector, as shown in the table below. The results show that our attack is also effective against Task Vector. Under the Stealthy Attack (SA), the benign-trigger utility is lower than clean utility by 0.115 (when removing 5%) and 0.095 (when removing 10%). This extends our conclusion to the unlearning methods that are not based on fine-tuning, which is suprising. We will include these findings in our revised version.
>
> |||Unlearn|Clearn Utility|Benign-trigger Utility|
> |-|-|-|-|-|
> |5%|SA|0.362|0.730|0.615|
> |10%|SA|0.380|0.735|0.640|
>
> (Experiments in rebuttal only use one seed and might be a little different from main paper.)
>
> **W2**: The tables and results are quite hard to read. There is no bolding, and decimal points are reported to three places. It took me a long time to parse the results.
>
> **A**: Thank you for pointing out this issue. We will use bold formatting to highlight the best results in our revision.
>
>
> **Q1**: How effective do you expect stealthy attack to be in the presence of more robust anomaly detection systems? It seems to me it might be easy to spot the specific attack. Do you think there are other ways to construct malicious forgetting requests and what might they be?
>
> **A**: To clarify, if model builders employ anomaly detection methods to identify our trigger (e.g., by checking whether certain common tokens appear with unusually high frequencies), they might be able to defend against our approach. However, there are other ways to construct poisoned forgetting data that are more stealthy. For instance, some common tokens are frequently used as names—such as Hope, Joy, Grant, and Will—so a high frequency of them would not necessarily raise suspicion. We conducted an experiment using *Will* as a benign trigger by submitting a request to remove information about *Will Hunting*. The model is unlearned by GD. We found that the benign-trigger utility dropped significantly from 0.57 to 0.1.
>
> |||Clean Utility|Benign-trigger Utility|
> |-|-|-|-|
> |GD|SA|0.570|0.100|
>
> **Q2**: While Scope-aware Unlearning shows promising results, the scope samples are constructed by concatenating forgetting and retaining prompts. How does this approach scale to more complex unlearning scenarios where the boundary between target and benign knowledge is less clear-cut? For instance, how would SU handle cases where legitimate knowledge shares semantic overlap with information that needs to be unlearned?
>
> **A**: Thank you for your question. Even if the distributions of forgetting data and retaining data are similar, the samples within each mini-batch during training are still different. For each mini-batch, the scope-aware term ensures that the presence of forgetting samples does not negatively affect *other samples*. These *other samples* do not necessarily have to be retaining data; they only need to be different from the samples used in the forgetting loss term.
>
> We validated this with an extreme experiment: we made the Scope-aware term's forgetting data samples and retaining data samples exactly the same from the perspective of dataset distribution by replacing the retaining samples with other forgetting samples. Specifically, we kept the original definitions of the forgetting loss term and retaining loss term unchanged. But for the Scope-aware term, instead of concatenating forgetting data with retaining data, we replaced the retaining data with randomly sampled forgetting samples from the forgetting set. In other words, we concatenated the forgetting samples (the same ones used in the forgetting loss) with other randomly sampled forgetting samples. In this setting, the forgetting data and benign data in the Scope-Aware term have 100% overlap from the perspective of dataset distribution, yet they remain different at the mini-batch level.
>
> The results are shown in the table below: we observed that the benign-trigger utility under attack remained nearly the same as clean utility. This indicates that even when there is overlap between the forgetting data and benign data in the Scope-Aware term, it can still defend against SA.
>
> |||Unlearn|Clean Utility|Benign-trigger Utility|
> |-|-|-|-|-|
> |GD|SU|0.044|0.500|0.497|
> |NPO|SU|0.239|0.713|0.679|
>
> **Q3**: The scope samples are constructed by simple concatenation - have you experimented with more sophisticated composition strategies, such as interleaving tokens? How does the choice of concatenation order affect performance?
>
> **A**: In our main paper, we concatenated the forgetting samples before the retaining samples. Here, we present experiments using the reversed order (forgetting samples after retaining samples) and interleaving tokens. We observe that the reversed order does not affect the benign-trigger utility. Instead, interleaving tokens impacts both the clean utility and benign-trigger utility. This is because interleaving forgetting and retaining tokens disrupts sentence structure, resulting in nonsensical sentences.
>
> ||||Unlearn|Clean Utility|Benign-trigger Utility|
> |-|-|-|-|-|-|
> |Reversed|GD|SU|0.016|0.516|0.467|
> ||NPO|SU|0.291|0.770|0.743|
> |Interleave|GD|SU|0|0.252|0.195|
> ||NPO|SU|0.211|0.611|0.595|
>
> **Q4**: Do you have any more general metrics of utility on standard language modeling benchmarks? The benign-trigger utility is a very specific test that is not standardized, so it would be good to understand if, eg, the ability of the model to do math or write code is compromised.
>
> **A**: Thank you for your question. We use "how many" as benign triggers and test our attack on GSM8K (Grade School Math 8K). This benchmark consists of math questions, some of which include "how many" in their prompts. We observe that SA reduces the performance from 0.130 to 0.111, while SU is able to recover it to 0.127. This provides a practical example of how math performance can be compromised without inserting benign triggers into the test set.
>
> ||Clean Utility|
> |-|-|
> |no|0.130|
> |SA|0.111|
> |SU|0.127|

---

### Decision · Program_Chairs · 2025-09-17

**Decision:**

Accept (poster)

**Comment:**

All reviewers unanimously agree that this paper should be accepted and I support their decision.

The paper shows a ``Stealthy Attack'' on fine-tuning–based LLM unlearning: by padding forget requests with common benign tokens (“please,” etc.), an adversary causes the model to overgeneralize the unlearning signal and start refusing or degrading normal queries. By pinpointing missing scope control and poor separation between benign and forget tokens as the root cause, the authors propose Scope-aware Unlearning. It adds a simple scope term with mixed “scope” samples to localize forgetting while preserving benign behavior.


Reviewers largely pressed on too-narrow evaluation (few models), missing baselines like inference-time unlearning (Task Vectors), no ablation of the scope coefficient, and thin attacks (MIA/relearning), and unclear novelty vs some recent  lines of work. The authors answered by adding results across more LLMs (Phi, Zephyr) and unlearning regimes (GD/NPO/IDK and Task Vectors), running an $\eta$-ablation and MIA/relearning tests, and clarifying threat model and differences to prior art—leading most reviewers to keep or raise positive scores.

Overall, reviewers were satisfied with the rebuttal and negative leaning reviewers updated to positive leaning.